# ChemHGNN: A Hierarchical Hypergraph Neural Network for Reaction Virtual Screening and Discovery

## Abstract

Reaction virtual screening and discovery are fundamental challenges in chemistry and material science, where traditional graph neural networks (GNNs) struggle to model multi-reactant interactions. In this work, we propose **ChemHGNN**, a hypergraph neural network (HGNN) framework that effectively captures high-order relationships in reaction networks. Unlike GNNs, which require constructing complete graphs for multi-reactant reactions, ChemHGNN naturally models multi-reactant reactions through hyperedges, enabling more expressive reaction representations. To address key challenges—such as combinatorial explosion, model collapse, and chemically invalid negative samples—we introduce a reaction center-aware negative sampling strategy (RCNS) and a hierarchical embedding approach combining molecule, reaction and hypergraph level features. Experiments on the USPTO dataset demonstrate that ChemHGNN significantly outperforms HGNN and GNN baselines, particularly in large-scale settings, while maintaining interpretability and chemical plausibility. Our work establishes HGNNs as a superior alternative to GNNs for reaction virtual screening and discovery, offering a chemically informed framework for accelerating reaction discovery.

## 1 Introduction

The discovery of chemical reactions is fundamental to advancements in fields ranging from drug development to materials science (Wang et al., 2023). With the advent of machine learning, data-driven methods have shown great promise in predicting potential chemical reactions, particularly by leveraging complex relational data (Coley et al., 2019; 2017; Yu et al., 2024). The combinatorial explosion of possible reactant combinations—stemming from over 60 million catalogued molecules (Ruddigkeit et al., 2012)—renders exhaustive experimental or computational enumeration infeasible. To address this challenge, reaction virtual screening offers a scalable in silico solution by evaluating sets of reactants and assigning a score that reflects the likelihood of a reaction occurring. This approach is critical for accelerating the reaction discovery process by narrowing down viable candidates efficiently.

Traditional graph-based models are well-developed to capture the bond changes within molecules in a specific reaction (Coley et al., 2019). However, traditional graph structures struggle to represent reaction networks involving multiple reactants, as these reactants interact with one another. Accurately modeling such interactions would require constructing a complete graph for each reaction, which becomes computationally impractical and conceptually inefficient. Hypergraphs generalize graphs to model these multi-way interactions as well as to capture high-level information within reaction networks besides the connectivity changes within a specified reaction. As such, hypergraphs offer a powerful framework for representing chemical reaction networks and discovering new reactions (Mann & Venkatasubramanian, 2023).

In this study, we propose a novel hypergraph neural network (HGNN) approach specifically designed for chemical reaction virtual screening and discovery. Our model exploits the ability of hypergraphs to naturally represent multi-reactant reactions, capturing the combinatorial complexity of chemical transformations. Chemical information is incorporated into the model through a hierarchical embedding strategy that integrates a reaction center-pretrained graph neural network and an HGNN to capture molecule, reaction, and hypergraph level details. A key challenge in training such models is the construction of effective negative samples which are typically absent in reaction datasets, but are nonetheless essential for optimizing performance and the prediction of chemically plausible

reactions. Additionally, efficiently generating new reactant combinations is critical for maximizing the likelihood of discovering novel reactions. To tackle these challenges, we introduce a tailored negative sampling strategy within the hypergraph framework, designed to differentiate valid chemical reactions from random or invalid combinations of reactants by introducing virtual nodes into the reaction hypergraph. Finally, we implement an efficient filtering mechanism to select viable reaction candidates based on the representations derived from our HGNN. **We summarize the key challenges as follows:**

- Combinatorial explosion of possible reactant combinations.
- Inadequacy of traditional graph models for multi-reactant reactions.
- Difficulty in generating effective negative samples for training.
- Need for chemically informed reaction representations.

**Our main contributions are as follows:**

- We find that HGNNs are better reaction representation learners and less prone to model collapse than traditional GNNs across dataset sizes.
- We propose ChemHGNN, an end-to-end pipeline for virtual reactant screening and reaction discovery that leverages hypergraphs as expressive reaction network representations.
- We introduce a domain-informed negative sampling method tailored for reaction learning, which is found to enhance model performance.
- Experiments show that ChemHGNN provides excellent performance at predicting reactive combinations of molecules. In particular, ChemHGNN displays the ability to extrapolate to reaction templates beyond those it was trained on, demonstrating an understanding of fundamental chemical reactivity.

This approach offers a chemically-informed framework for reaction virtual screening and discovery, with broad implications for molecular design and accelerated materials discovery.

## 2 PRELIMINARIES

### 2.1 PROBLEM FORMULATION

Let a hypergraph reaction network be $\mathcal{H} = (\mathcal{V}, \mathcal{E})$, where $\mathcal{V}$ represents a set of reactants, $\mathcal{E}$ is a set of reactions, and $\boldsymbol{X}$ is the set of node features. Reaction virtual screening can be formulated as a predictor $f(\cdot)$ which takes a set of reactants $\{v_i \mid v_i \subseteq \mathcal{V}\}$, and outputs a score representing the likelihood of a reaction to happen within $\{v_i\}$.

$$x = [\cdots \boldsymbol{X}_i \cdots], f(x) = k \in \mathbb{R} \qquad (1)$$

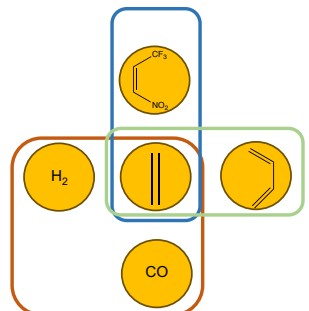

### 2.2 CONSTRUCTION OF THE HYPERGRAPH

A hypergraph is an ordered pair $\mathcal{H} = (\mathcal{V}, \mathcal{E})$, where $\mathcal{V}$ is a set of nodes, and $\mathcal{E}$ is a set of hyperedges. Each hyperedge is a non-empty subset of nodes. The structure of a hypergraph is usually represented by an incidence matrix $\boldsymbol{H} \in \{0,1\}^{|\mathcal{V}| \times |\mathcal{E}|}$, with each entry $H_{v,e}$ indicating whether the vertex $v$ is in the hyperedge $e$ (Antelmi et al., 2023). We construct the hypergraph reaction network by representing molecules as nodes and reactions as hyperedges that

Figure 1: This hypergraph involves the reactants of two Diels Alder reactions and one Hydroformylation. Each reactant is presented as node, and colored outlines as the hyperedges.

enclose the nodes. Since virtual screening is performed on reactants without knowledge of the products, we exclude the products from the reactions and only focus on reactants, as illustrated in Fig. 1.

### 2.3 HYPERGRAPH NEURAL NETWORKS

We have defined incidence matrix $\boldsymbol{H} \in \{0,1\}^{|\mathcal{V}| \times |\mathcal{E}|}$, with entries defined as

$$H_{v,e} = \begin{cases} 1, & \text{if } v \in e, \\ 0, & \text{if } v \notin e. \end{cases} \qquad (2)$$

For a vertex $v \in \mathcal{V}$, its degree is defined as $d(v) = \sum_{e \in \mathcal{E}} w_e H_{v,e}$, where $w_e$ is the weight of the edge. For an edge $e \in \mathcal{E}$, its degree is defined as $\delta(e) = \sum_{v \in \mathcal{V}} H_{v,e}$. Further, $\boldsymbol{D}_e$ and $\boldsymbol{D}_v$ denote the diagonal matrices of the edge degrees and the vertex degrees, respectively.

When we have a hypergraph signal $\boldsymbol{X} \in \mathbb{R}^{|\mathcal{V}| \times C_1}$ with $|\mathcal{V}|$ nodes and $C_1$ dimensional features, our hyperedge convolution can be formulated by

$$Y = D_v^{-1/2} HW D_e^{-1} H^\top D_v^{-\frac{1}{2}} X\Theta, \tag{3}$$

(Feng et al., 2019) where $\boldsymbol{W}$ is initialized as identity matrix, meaning equal weights for all hyperedges. $\Theta \in \mathbb{R}^{C_1 \times C_2}$ is the parameter to be learned during the training process. The filter $\Theta$ is applied over the nodes in hypergraph to extract features. After convolution, we can obtain $\boldsymbol{Y} \in \mathbb{R}^{|\mathcal{V}| \times C_2}$, which can be used for further downstream tasks.

## 2.4 Weisfeiler-Lehman Network for Reaction Center Prediction

The Weisfeiler-Lehman Network (WLN) (Jin et al., 2017) is a powerful graph-based architecture inspired by the Weisfeiler-Lehman graph isomorphism test. It captures structural features of molecular graphs by iteratively updating node embeddings through message passing. In the context of hypergraph-based reaction networks, we can project the hypergraph into a molecular graph and apply WLN to refine node representations for reaction center prediction.

Let $\mathcal{G} = (\mathcal{V}, \mathcal{E}_g)$ be the projected molecular graph where $\mathcal{V}$ are the atoms (or reactants) and $\mathcal{E}_g$ are the chemical bonds (or interaction edges). Each node $v \in \mathcal{V}$ is associated with a feature vector $\boldsymbol{h}_v^{(0)} \in \mathbb{R}^d$, where $d$ is the dimensionality of atom features.

The WLN updates the node representations in $L$ layers using neighborhood aggregation:

$$\boldsymbol{m}_v^{(l)} = \sum_{u \in \mathcal{N}(v)} \phi\left(\boldsymbol{h}_v^{(l-1)}, \boldsymbol{h}_u^{(l-1)}, \boldsymbol{e}_{uv}\right), \tag{4}$$

$$\boldsymbol{h}_v^{(l)} = \psi\left(\boldsymbol{h}_v^{(l-1)}, \boldsymbol{m}_v^{(l)}\right), \tag{5}$$

where $\mathcal{N}(v)$ denotes the neighbors of node $v$, $\boldsymbol{e}_{uv}$ is the edge feature between $u$ and $v$, and $\phi, \psi$ are neural network functions (e.g., MLPs or GRUs). The final representation $\boldsymbol{h}_v^{(L)}$ captures both local and neighborhood information up to $L$ hops.

To predict the likelihood of a node $v$ being a reaction center, a readout function is applied over the final node embedding:

$$\hat{y}_v = \sigma(\boldsymbol{W}_o \boldsymbol{h}_v^{(L)} + \boldsymbol{b}_o), \tag{6}$$

where $\boldsymbol{W}_o$ and $\boldsymbol{b}_o$ are trainable parameters and $\sigma$ is the sigmoid activation function. This outputs a score $\hat{y}_v \in [0, 1]$ indicating the probability of $v$ participating in the reaction center.

## 2.5 Construction of Cliques from Hyperedges

In graph theory, the concept of a clique plays a fundamental role in understanding dense substructures within a graph. A *clique* is a subset of vertices of an undirected graph such that every two distinct vertices are adjacent. More formally, a clique of size $k$ in a graph $\mathcal{G} = (\mathcal{V}, \mathcal{E})$ is a set of vertices $\{v_1, v_2, \ldots, v_k\} \subseteq \mathcal{V}$ such that $(v_i, v_j) \in \mathcal{E}$ for all $1 \le i < j \le k$.

The construction of a clique from a hypergraph involves interpreting each hyperedge as a complete subgraph among its constituent vertices. That is, given a hypergraph $\mathcal{H} = (\mathcal{V}, \mathcal{E})$, we can define an associated graph $\mathcal{G} = (\mathcal{V}, \mathcal{E})$ where an edge $(u, v) \in \mathcal{E}$ exists if and only if there exists a hyperedge $e \in \mathcal{E}$ such that $(u, v) \in e$. This process is commonly referred to as the *2-section* or *clique expansion* of a hypergraph. We employ clique expansion to compare the performance of an HGNN to a variety of GNN baseline models.

## 3 Methodology

We present our framework of ChemHGNN in Fig. 2. Given a reaction network defined by a reaction dataset, the negative sampling blocks generate negative hyperedges for training. The model learns molecular-, reaction- and hypergraph-level representations for molecules in the reaction network through the attention mechanism. The final pooling module provides a score for the combination of the molecules input as a potential reaction.

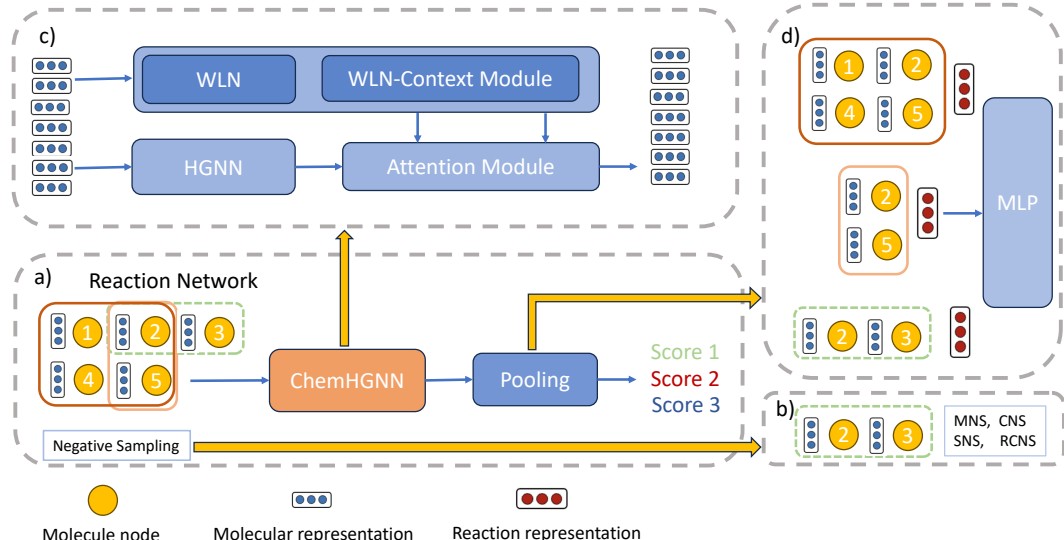

Figure 2: a) The ChemHGNN pipeline. b) is the negative sampling block which involves different strategies for generating negative samples within reaction network. In c), the ChemHGNN can be broken into HGNN module, WLN module and an attention module that learns the representation from the molecular level, reaction level and hypergraph level with cross-attention. d) is an MLP layer with a pooling mechanism for merging the representation of the molecules.

## 3.1 A SPECIALIZED HYPERGRAPH NEURAL NETWORK FOR CHEMISTRY DOMAIN

We propose a new HGNN model with a GNN pretrained on the reaction center prediction task to enhance its internal understanding of molecular reactivity while maintaining its capability to capture the high-order relationship between molecules.

The WLN module in Fig 2 c) was pretrained on the USPTO-410k dataset to enable the model to learn intermolecular bond changes at the reaction level.

The model convolutes original features $\boldsymbol{X}^{(0)}$ of the nodes from ECFP6 encodings with a two-layer HGNN

$$\boldsymbol{X}^{(l+1)} = \sigma\left(\boldsymbol{D}_v^{-1/2}\boldsymbol{H}\boldsymbol{W}\boldsymbol{D}_e^{-1}\boldsymbol{H}^\top\boldsymbol{D}_v^{-\frac{1}{2}}\boldsymbol{X}^{(l)}\boldsymbol{\Theta}^{(l)}\right) \tag{7}$$

The output embedding $\boldsymbol{X}$ as well as the reaction embedding $\boldsymbol{X}_{\mathrm{GNN}}$ from the GNN pretrained on reaction center classification are attended with the attention module in order to correlate molecular level information with reaction and hypergraph level information.

$$\mathrm{Attention}\left(\boldsymbol{W}_Q \cdot \boldsymbol{X}, \boldsymbol{W}_K \cdot \boldsymbol{X}_{\mathrm{GNN}}, \boldsymbol{W}_V \cdot \boldsymbol{X}_{\mathrm{GNN}}\right) = \mathrm{softmax}\left(\frac{\boldsymbol{Q}\boldsymbol{K}^\top}{\sqrt{d_k}}\right)\boldsymbol{V} = \boldsymbol{X}_{\mathrm{attn}} \tag{8}$$

The output embeddings of the HGNN nodes together with the output embedding from attention module are concatenated and go through an aggregator(SUM) to retrieve the final embedding of the reactions.

$$\boldsymbol{X}_{\mathrm{react}} = \mathrm{AGGREGATE}((\boldsymbol{X}||\boldsymbol{X}_{\mathrm{attn}})) \tag{9}$$

This is then sent through an MLP to obtain the final scores of the reaction occurence.

$$\hat{y} = \mathrm{MLP}(\boldsymbol{X}_{\mathrm{react}}) \tag{10}$$

We consider two losses in this model. The first one is binary cross entropy loss between the predicted and true label:

$$\mathcal{L}_{\mathrm{BCE}} = -\frac{1}{N}\sum_{i=1}^{N}\left[y_i\log\hat{y}_i + (1-y_i)\log(1-\hat{y}_i)\right] \tag{11}$$

The second loss is the mean squared error (MSE) between the sum of the molecular embeddings and $\vec{0}$. Through $\boldsymbol{X}_{GNN}$, bond change information is encoded into the molecular embeddings. Bond changes are usually complementary within a reaction; that is, when a bond is broken in a specific location, it is likely to form in another location. Therefore, we expect the sum of the molecular embeddings to be close to $\vec{0}$, resulting inthe following MSE loss:

$$\mathcal{L}_{\text{MSE}} = \frac{1}{N} \sum_{i=1}^{N} \|r_i - \mathbf{0}\|^2 = \frac{1}{N} \sum_{i=1}^{N} \|r_i\|^2 \tag{12}$$

### 3.2 Reaction Center Negative Sampling

Various sampling techniques have been proposed to generate negative samples in hypergraphs. In this study, we utilize three representative methods—Motif Negative Sampling (MNS), Clique Negative Sampling (CNS), and Sized Negative Sampling (SNS) (Hwang et al., 2022)—to generate negative samples in the reaction hypergraph. Additionally, we introduce a novel strategy, **Reaction Center Negative Sampling (RCNS)**, which combines a stochastic approach with reaction center information.

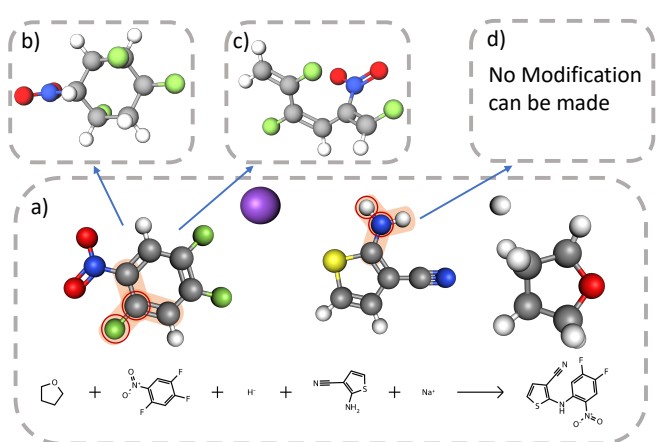

Figure 3: This is a visualization of RCNS. a) identifies the reaction center atoms circled in red and the bond associated with the reaction center highlighted in orange. In b), the aromaticity of 1,2,4-trifluoro-5-nitrobenzene is eliminated. In c), there is C-C bond cleavage in the benzene ring. In d), any modification of the reaction center of 2-aminothiophene-3-carbonitrile will result in an invalid molecule

Reactivity is eliminated in RCNS by selectively perturbing the key structural elements responsible for the chemical transformation. Specifically, the method first identifies the reaction center atoms—those directly involved in bond-making or bond-breaking events—within a reaction hyperedge. It then modifies or removes bonds connected to these atoms, disrupting the critical molecular topology necessary for the reaction to occur. For instance, in the Figure 3, destroying a ring or breaking a functional linkage prevents the reactants from forming the intended product, thereby creating a chemically plausible but non-reactive negative sample. These altered reactants are subsequently incorporated as virtual nodes in the hypergraph, and a corresponding hyperedge is added to support model training on negative samples.

### 3.3 Sort Out Block

We employ simulated annealing (SA) in a final sort out block to enable efficient search for promising candidate reactions based on the scores returned by ChemHGNN (Algorithm 2). SA is a probabilistic optimization technique inspired by the annealing process in metallurgy (Chopard & Tomassini, 2018). It is particularly effective for solving complex optimization problems where the search space is large and contains numerous local optima.

In our sort out mechanism, we randomly select a molecule as initial solution $s$ and calculate the change in the objective function. Here we use the Euclidean norm of sum of vectors as the objective function to measure solution quality:

$$f(\mathbf{x}) = \|\boldsymbol{x}\|_2 = \left( \sum_{i=1}^{n} x_i^2 \right)^{\frac{1}{2}} \tag{13}$$

Where:

- $f : \mathbb{R}^n \to \mathbb{R}$ is the objective function.
- $x \in \mathbb{R}^n$ represents the sum of the input vector (referred to as **molecular representation**).
- $\| \cdot \|_2$ denotes the Euclidean norm or $\ell_2$ norm.

Subsequent molecules are subjected to an acceptance mechanism. If the addition of the molecule displays improvement over the last iteration, the change is accepted. If not, there is still possibility to accept the change, tuned by a control temperature that allows a balance between exploration and exploitation. The process is repeated until the maximum number of molecules to select in a reaction is reached.

## 4 EXPERIMENT

### 4.1 DATASET

The datasets for the following experiments are subsets of the USPTO-410k dataset (Lowe, 2017). We randomly curate datasets of 1k, 5k, and 10k datapoints to assess the performance of model at different scales.

### 4.2 HGNN IS A BETTER REACTION REPRESENTATION LEARNER THAN GNN

**Setup:** A baseline HGNN model and several GNN baselines (GCN, NOCD, GAT) were trained on datasets of scale 1k, 5k and 10k. A 1:1 NS ratio in the training sets were achieved through an even mixture of SNS, MNS, CNS, and RCNS. Model performance was evaluated on positive samples and negative data generated from various NS strategies. To demonstrate performance trends, we depict the SNS strategy in our visualizations, see Fig. 4. Full performance metrics for all NS strategies can be found in Appendix Table 5:

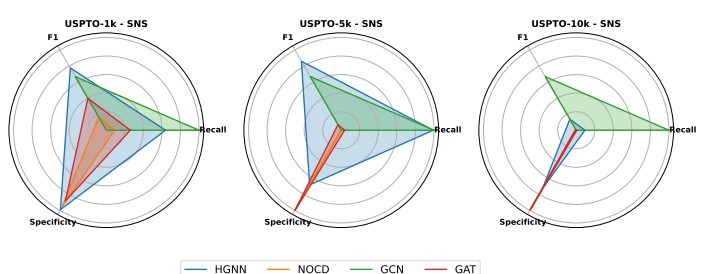

Figure 4: The radar plots compare the performances of the baseline GNN models against a baseline HGNN on datasets of different scales and tested against the SNS strategy. The extremely thin shaded regions for certain models (e.g., NOCD, GAT) highlight sharp metric imbalance, perfect specificity with near-zero recall. The green shaded area (e.g., GCN) highlights a high recall, near-zero specificity region, both indicating model collapse.

(1) **Graph Neural Network-based models suffer from model collapse even on small scale datasets.** In USPTO-1k, the GCN starts to collapse, where we observe specificity to be 0 in all of the NS strategies. The phenomenon expands to NOCD with specificity almost 1, with collapse worsening as dataset size increases.

(2) **HGNN greatly outperforms the GNN based models if it does not collapse.**

Excluding GCN, which suffers from model collapse, in the USPTO-1000, the HGNN has an overwhelming improvement over the GNN-based models on all of the NS Strategies on F1 score.

(3) **Both HGNN and GNNs suffer from serious model collapse when the scale of the dataset goes up.** In USPTO-10k, we can see a clear pattern that HGNN is collapsing or approaching collapse, e.g. a recall of 0.0938 at USPTO-10k with SNS strategy. Interestingly, even though the baseline HGNN outperforms the GNN in small dataset. The pure intermolecular information is not sufficient for the hypergraph to capture reactivity when the scale of the reaction network expands.

**Some discussion:** In the traditional graph, complete graphs are typically needed to connect all reactants. However, the question of expressing a reaction is simplified in a hypergraph. A reaction could be enclosed in just one hyperedge. The limitation of traditional graphs in the expression of a reaction network extends to the redundant connections between reactants, where the connection only

informs that the reactants have shared reactions, while not informing what the shared reactions are, and what roles the reactants play. For this reason, the representation learned from a GNN may fail to capture the real connectivity and relationships between molecules in a reaction network. The greater expressiveness of the hypergraph may allow the HGNN to better capture reactivity and the high order relationships between molecules. Moreover, the information aggregated from the GNN deviates from the real reaction. The weight per neighbor being aggregated is vertex degree-based, which causes it to only receive partial reaction-level information. HGNNs aggregate information both on the degree of the edge and the vertex, providing more complete reaction-level information. For instance, when a reactant participates in multiple reactions, it tends to aggregate more information from reactions involving fewer co-reactants.

### 4.3 CHEMHGNN OUTPERFORMS HGNN AND GNN BASELINES

To address the issue of HGNN and GNN collapse when the scale of reaction network expands, we propose ChemHGNN, which incorporates domain knowledge at the molecule, reaction, and hypergraph level to predict the likelihood of a reaction to occur.

**Setup:** We compared ChemHGNN with the baseline model HGNN and the NOCD (GNN) trained under a mixture of negative sampling strategies. We will focus on the evaluation metrics arising from testing on the SNS strategy for simplicity, see Fig. 5. More information can be found at Appendix Table 6, 7, 8. We have the following observations:

(1) **ChemHGNN outperforms HGNN and GNN baselines, especially at larger dataset sizes.** On metrics of F1 score, accuracy, and recall, ChemHGNN outperforms the baseline models in the majority of cases, with the effect becoming most pronounced on USPTO-10k.

(2) **At all dataset sizes, the ChemHGNN is less prone to model collapse.** As observed previously, even at the scale of 1k datapoints, GNNs tended to-

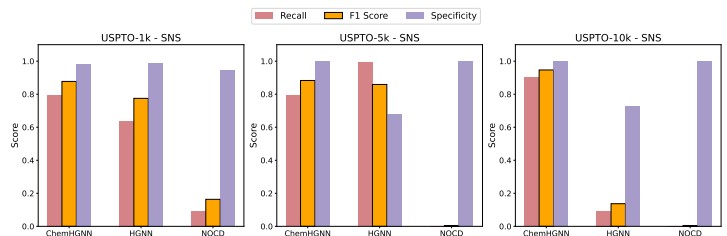

Figure 5: The chart describes the performance of the **ChemHGNN** and baseline models (HGNN and NOCD) trained on datasets of different scales and tested on SNS strategy. Due to model collapse, some of the bars are **invisible** (e.g., in NOCD). Among the plotted evaluation metrics, the F1 Score is **highlighted** using a distinct yellow bar with a black border to draw attention to its importance, particularly relevant in evaluating model performance under class imbalance or skewed data.

wards model collapse. As we scale up the dataset, the model collapse of GNN becomes more dominant, with less than ten testing datapoints out of a thousand being predicted as positive. In contrast, ChemHGNN at all dataset sizes is well-balanced in its predictions. This opens up the ability to enlarge the virtual screening database used for training, building a more powerful model with the ChemHGNN architecture.

(3) **ChemHGNN sees improved performance as the size of the datasets scales up.** The trend in Fig 5 shows that the F1 score of the ChemHGNN increases from 0.87 to 0.94 as the dataset scales up from 1k to 10k, suggesting that increasing the dataset sizes even further may yield more powerful models.

To better understand how the behavior of the model correlates with the diversity of reactions in the training dataset, we investigated factors related to the reaction types present in the dataset.

**Setup:** The USPTO training reactions were classified using rxnfp to 1000 templates, and datasets of 10k datapoints were curated from the top 3 most frequently-occurring reaction templates (RT 274, RT 586, RT 672) (Schwaller et al., 2021). Details of the reaction template distribution in the USPTO-10K dataset can be found in the appendix (Fig. 11 b). We have the following observations:

(4) **GNN performance improves when predicting a single reaction template, but is still generally outperformed by ChemHGNN,** seen in Table 13. When subjected to the easier task of predicting reactivity of a single reaction template, GNN performance improves and models are less prone to

collapse. However, ChemHGNN still outperforms GNNs on this task, with the exception of a few cases on RT 672.

We also curate three additional datasets from USPTO-10k by leaving one reaction template out to serve as the testing dataset to investigate the extrapolation capability of the model. We have the following observations:

(5) **The generalization of the model on unseen reaction templates shows promising performance,** see Fig. 6. ChemHGNN trained on USPTO-10k dataset identifies 87.02% of reaction template 672 (Partial Reduction of Pyrrole to Pyrroline), 72.92% of reaction template 274 (Friedel–Crafts Acylation) and 80.62% of reaction template 586 (Grignard Addition).

### 4.4 HGNNs PROVIDE REACTION REPRESENTATIONS THAT CAN BETTER SEPARATE REACTIVE AND UNREACTIVE REACTANT COMBINATIONS.

To better understand the differences in model performance between the NOCD, HGNN and ChemHGNN models, we plot the 2D and 3D t-SNE. Here we focus on 2D t-SNE of GNN, HGNN and ChemHGNN trained on USPTO-1k, see Fig. 7. More 2D and 3D tSNE plots can be found in Appendix Fig. 14 and Fig. 15.

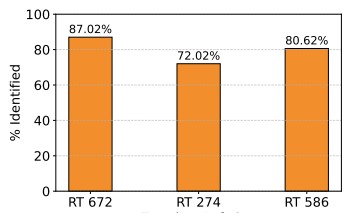

(1) **HGNN provides tighter clusters than the GNN baseline** In the baseline HGNN reaction representation, we observe distinct clustering between representations, in contrast to the diffuse distribution of the GNN. However, even though the HGNN baseline model representations are better clustered, there is only one clear region where true and false labels are well separated. Regardless, this is a marked improvement over the GNN, where there is little distinction between labels.

(2) **The ChemHGNN reaction representations display separation between true and false labels.** In Fig. 7, even though the reaction representations from the ChemHGNN model are not as tightly clustered as the baseline HGNN model, it has a clearer high-dimensional boundary separating the true and false labels.

Figure 6: In this bar plot, models are trained on a mixture of NS strategies as well as the USPTO-10k dataset with a specific reaction template excluded. Models are then tested on the excluded template to investigate the generalization of the model on unseen reactions.

### 4.5 SIMULATED ANNEALING ASSISTS IN EFFICIENT SELECTION OF HIGH-SCORING CANDIDATE REACTIONS

**Setup:** We use the molecular representations learned from the USPTO-1k, 5k, 10k to virtual screen possible reactant combinations using Simulated Annealing and Random Selection, and score them with the MLP modules. We have the following observations:

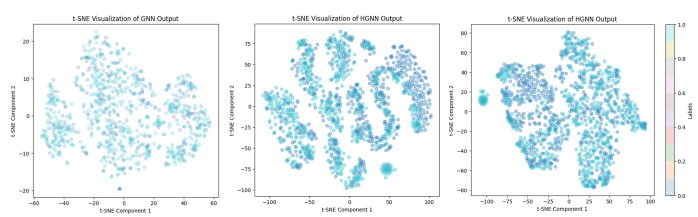

Figure 7: t-SNE comparison between baseline NOCD (GNN) model (left), baseline HGNN (mid) and ChemHGNN (right) on USPTO-1k

(8) **The best scores of combinations grow as the number of iterations grow.** Moreover, ChemHGNN trained on larger datasets typically provides better-scoring combinations than the smaller models. As seen in the Table 14 a), we can see the best result of ChemHGNN trained on USPTO-5k is better than USPTO-1k. However, for the larger USPTO-10k dataset, performance increased further as the number of iterations increases to 5M. This is partially due to a larger search space for USPTO-10k dataset and therefore requires more iterations to achieve a better score than USPTO-5k.

(9) **Simulated annealing (SA) can better target suitable reactant combinations than random selection.** In table 14 b), SA outperforms random selection at all numbers of iterations.

### 4.6 RCNS PROVIDES IMPROVEMENTS IN MODEL SPECIFICITY AND F1 SCORE

**Setup:** Since RCNS involves the creation of virtual nodes, we created two hypergraph reaction networks, one with the virtual nodes from RCNS, and the other without. We compared the ChemHGNN trained on the two different hypergraph reaction networks with mixed NS including and excluding RCNS. Models were tested on MNS, CNS and SNS negative samples.

(1) **The RCNS template distribution follows the distribution of the positive samples.** In Fig 11, both the generated negative samples and the positive samples peak at the template 672. The second highest is template 586 across both distributions, and the following peaks are also sub-linearly correlated to each other. However, there are no strict sub-linear correlations between the positive samples and negative samples as positive samples are not guaranteed to generate negative samples.

(2) **RCNS leads to positive gains in larger datasets and minor negative effects in small datasets, see Fig. 8.** For USPTO-5k and 10k, RCNS leads to performance improvements of +0.58%, and +7.75%, respectively. This indicates that RCNS scales well and benefits from larger data volumes, potentially due to better representation learning. On the USPTO-1k dataset, the inclusion of RCNS slightly degrades performance (-0.55% improvement), suggesting RCNS may introduce noise on smaller datasets.

(3) **RCNS enhances specificity and F1 score.** When RCNS is used, specificity and F1 scores tend to increase, especially when tested on the SNS strategy. For instance, in USPTO-10k with SNS, specificity jumps from 0.9580 to 0.9984, and F1 from 0.9274 to 0.9468, indicating a better balance between specificity and recall.

### 4.7 ABLATION STUDIES

Table 18 evaluates the performance of ChemHGNN on the USPTO-10k dataset under different ablation settings. Specifically, it examines how removing different architectural components (WLN, SUM, MSE) and applying different NS strategies affects metrics like Accuracy, Precision, Recall, F1 Score, Specificity, and decrement in F1 Score.

Figure 8: Comparison of **ChemHGNN** performance trained under different negative sampling (NS) strategies **with or without** reaction center-aware negative sampling (RCNS) across three dataset scales.

Removal of MSE loss results in a slight drop in F1 across all sampling methods (e.g., SNS drops about 1.41%). MSE seems important but not critical—its removal hurts performance modestly. Removal of SUM aggregator and MSE loss has a major performance degradation, especially in recall, suggesting that the SUM component is essential to properly aggregate information. Removing all three components, including WLN module has a drastic degradation of model performance. F1 scores plunges (e.g., 0.1371 for SNS, an 85.5% decrement).

## 5 CONCLUSION

In this work, we demonstrated that hypergraph neural network (HGNN) outperforms traditional graph neural networks (GNNs) in learning representations for chemical reaction networks, particularly in capturing multi-reactant interactions and high-order relationships. Our proposed ChemHGNN framework addresses key challenges, including combinatorial explosion, model collapse in large-scale datasets, and the need for chemically informed negative sampling. By integrating molecular, reaction and hypergraph level embeddings, and a novel negative sampling strategy (RCNS), ChemHGNN achieves robust performance across diverse reaction templates and scales. Experimental results validate its superiority over HGNN and GNN baselines, with improved specificity, F1 scores, and representation quality. This work advances reaction virtual screening and discovery by providing a scalable, chemically grounded framework, paving the way for accelerated discovery in drug development and materials science.

## 6 ETHICS STATEMENT

This work does not involve human subjects, sensitive personal data, or applications with foreseeable harmful consequences. All experiments are conducted on publicly available datasets, and we adhere strictly to the ICLR Code of Ethics. No conflicts of interest or ethical concerns arise from this study.

## 7 REPRODUCIBILITY STATEMENT

We have taken several steps to ensure reproducibility of our results. Detailed descriptions of the models, datasets, and negative-sampling strategies are provided in the main paper. Full implementation details, hyperparameter settings, and experiment scripts are available in the appendix. An anonymous repository containing the complete source code and instructions for reproducing all experiments has been made available at: `https://anonymous.4open.science/r/Hypergraph_Reaction_Discovery-B3C1/README.md`.

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

# A   APPENDIX / SUPPLEMENTAL MATERIAL

## CONTENTS

## A.1 NOTATION SUMMARY

Table 1: Summary of Notation

| Symbol | Description |
|---|---|
| $\mathcal{V}$ | Set of nodes (molecules) in the reaction network |
| $\mathcal{E}$ | Set of hyperedges (reactions) in the reaction network |
| $G = (\mathcal{V}, \mathcal{E})$ | Original reaction hypergraph |
| $G' = (\mathcal{V}', \mathcal{E}')$ | Augmented reaction hypergraph with negative samples |
| $e_i$ | A reaction (hyperedge) in the hypergraph |
| $v_i$ | A molecule (node) in the hypergraph |
| $v_i^-$ | Modified non-reactive molecule (virtual node) |
| $e_i^-$ | Negative sample hyperedge generated via RCNS |
| $C_i$ | Reaction center atoms for reaction $e_i$ |
| $\boldsymbol{X}^{(l)}$ | Node feature matrix at layer $l$ |
| $\boldsymbol{X}$ | Final output embedding from HGNN |
| $\boldsymbol{X}_{\text{GNN}}$ | Reaction embedding from pretrained GNN |
| $\boldsymbol{X}_{\text{attn}}$ | Attention-based embedding output |
| $\boldsymbol{X}_{\text{react}}$ | Final aggregated reaction embedding |
| $\boldsymbol{W}, \boldsymbol{D}_v, \boldsymbol{D}_e, \boldsymbol{H}$ | Weight matrix and structural matrices in HGNN |
| $\boldsymbol{W}_Q, \boldsymbol{W}_K, \boldsymbol{W}_V$ | Query, key, value projection matrices in attention |
| $\hat{y}$ | Predicted score for a reaction |
| $y_i$ | Ground truth label for reaction $i$ |
| $r_i$ | Bond change vector for molecule $i$ |
| $\mathcal{L}_{\text{BCE}}$ | Binary Cross Entropy loss |
| $\mathcal{L}_{\text{MSE}}$ | Mean Squared Error loss (zero-sum constraint) |
| $f(\boldsymbol{x})$ | Objective function in sort-out block |
| $\boldsymbol{x}$ | Sum of molecular representations in a candidate solution |
| $T$ | Temperature in Simulated Annealing |
| $\alpha$ | Cooling rate in Simulated Annealing |
| $L$ | Number of iterations per temperature level |
| $s, s'$ | Current and candidate solutions in Simulated Annealing |
| $\Delta E$ | Change in objective function |

## A.2 RELATED WORK

**Graph Neural Networks for Chemistry.** Graph Neural Networks (GNNs) have emerged as powerful frameworks for molecular representation learning since molecules naturally form graph structures with atoms as nodes and bonds as edges (Guo et al., 2023; Ma et al., 2024). The Message Passing Neural Network framework has become foundational, where atom embeddings are iteratively updated through information exchange with neighboring atoms (Gilmer et al., 2017). Despite their success, traditional GNNs face significant limitations when modeling multi-reactant systems, as they struggle to represent important chemical features including functional groups, spatial relationships, and multi-way interactions (Coley et al., 2019; Rong et al., 2020).

**Hypergraph Representation of Reaction Networks.** Conventional graphs can only represent pair-wise correlations, which is insufficient for modeling complex chemical systems where higher-order relationships are prevalent (Chen & Schwaller, 2024). Hypergraphs extend graphs to accommodate high-order correlations, making them especially suitable for representing multi-participant chemical reactions (Chang, 2024). Recent work has introduced chemical hypergraphs as a unified mathematical structure where metabolites serve as nodes and hybrid edges represent both metabolic directionality and high-order interactions (Huang et al., 2025). This approach more naturally captures the inherent higher-order relationships in reaction networks, including reactant-product directionality (Yadati et al., 2020).

**Hyperlink Prediction.** As a natural extension of link prediction on graphs, hyperlink prediction aims to infer missing hyperlinks in hypergraphs (Chen & Liu, 2023). This has direct applications in chemical reaction networks, where methods can be categorized into similarity-based, probability-based, matrix optimization-based, and deep learning-based approaches. Neural Hyperlink Predictor (NHP) adapts GCNs for link prediction in both undirected and directed hypergraphs, with NHP-D being the first method specifically designed for directed hypergraphs (Yadati et al., 2020). The hypergraph representation preserves reaction context and uncovers hidden insights not apparent in traditional directed graph representations (Mann & Venkatasubramanian, 2023). Hypergraph Neural Networks have demonstrated superior performance in molecular property prediction tasks, even outperforming models that utilize 3D geometric information (Chen & Schwaller, 2024).

## A.3 NS STRATEGY OVERVIEW

- **Motif Negative Sampling (MNS)**

    - Fills a set of size $K$ through **clique expansion**.

- **Clique Negative Sampling (CNS)**

    - Picks a **random hyperedge**.
    - Replaces a **random node** that is adjacent to all other constituent nodes.

- **Sized Negative Sampling (SNS)**

    - Fills a set with $K$ **random nodes**.
    - This is the most naive way of generating negative hyperedges.

- **Reaction Center Negative Sampling (RCNS)**

    - Identifies the **reaction center**(atoms) in the reaction.
    - Edits the bond surrounding those reaction centers to eliminate the reactivities between reactants

The distribution of the template in RCNS follows the distribution of the original distribution of template in the reaction network.

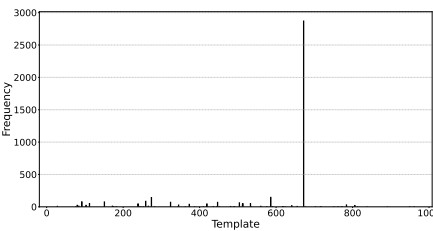 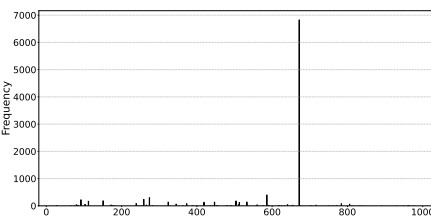

Figure 9: *

**(a)** RCNS reaction type distribution
(USPTO-10k)

Figure 10: *

**(b)** Original reaction type distribution
(USPTO-10k)

Figure 11: Comparison of reaction template distributions between RCNS-generated samples from USPTO-10k and the original USPTO-10k dataset.

### A.4 ALGORITHM

#### A.4.1 RCNS ALGORITHM

---
**Algorithm 1** Reaction Center Negative Sampling (RCNS)

---
1: **Input:** Reaction Network $\mathcal{G} = (\mathcal{V}, \mathcal{E})$, where $\mathcal{V} = \{v_1, v_2, ..., v_n\}$ and $\mathcal{E} = \{e_1, e_2, ..., e_n\}$each $e_i$ is a hyperedge (reaction)
2: **Output:** Augmented Reaction Network $\mathcal{G}' = (\mathcal{V}', \mathcal{E}')$ with negative samples
3: **for** each reaction $e_i \in \mathcal{E}$ **do**
4:      Identify **reaction center atoms** $C_i$ using set operations on reactants and products
5:      **Modify bonds** surrounding $C_i$ to destroy reactivity
6:      Generate **non-reactive reactants** $v_i^-$ (e.g., by breaking ring structures)
7:      Create **virtual nodes** in the hypergraph for $v_i^-$
8:      Add new **negative hyperedge** $e_i^-$ with replacing $v_i$ with $v_i^-$ to the hypergraph
9: **end for**
10: **return** Augmented hypergraph with added negative samples

---

#### A.4.2 SIMULATED ANNEALING ALGORITHM

---
**Algorithm 2** Simulated Annealing Algorithm

---
1: **Input:** Objective function $f(s)$, initial solution $s$, initial temperature $T$, cooling rate $\alpha$, number of iterations $L$
2: **Output:** Approximate optimal solution
3: **while** stopping criterion not met **do**
4:      **for** $i \leftarrow 1$ to $L$ **do**
5:          Generate a neighboring solution $s'$
6:          Calculate the change in objective function: $\Delta E \leftarrow f(s') - f(s)$
7:          **if** $\Delta E < 0$ **then**
8:              Accept $s'$ as the new solution: $s \leftarrow s'$
9:          **else**
10:              Accept $s'$ with probability $P \leftarrow \exp\left(-\frac{\Delta E}{T}\right)$
11:              **if** random number in $[0, 1] < P$ **then**
12:                  $s \leftarrow s'$
13:              **end if**
14:          **end if**
15:      **end for**
16:      Reduce the temperature: $T \leftarrow \alpha \cdot T$
17: **end while**
18: **Return** the best solution found

---

## A.5  Detailed Experimental Setup

### A.5.1  Training, Testing, Validation set selection

For all USPTO datasets (USPTO-1k, USPTO-5k, USPTO-10k), the training, validation, and test splits were selected randomly from the full dataset to ensure representative coverage of reaction types. Within each split, positive and negative samples were balanced such that the ratio of positive to negative samples (with 1/4 from each negative sampling strategy) is approximately 1:1 during both training and testing. This design helps mitigate class imbalance and ensures that the model receives sufficient examples of both outcomes for effective learning and evaluation.

Table 2: Train/Validation/Test splits for USPTO datasets. Positive (Pos) and Negative (Neg) counts are reported.

| Dataset | Train | | | Validation | | | Test | | |
|---|---|---|---|---|---|---|---|---|---|
| | Pos | Neg | Total | Pos | Neg | Total | Pos | Neg | Total |
| USPTO-1k | 579 | 499 | 1,078 | 192 | 166 | 358 | 194 | 166 | 360 |
| USPTO-5k | 2,863 | 2,500 | 5,363 | 954 | 832 | 1,786 | 955 | 833 | 1,788 |
| USPTO-10k | 5,721 | 4,995 | 10,716 | 1,906 | 1,662 | 3,568 | 1,908 | 1,666 | 3,574 |

### A.5.2  Setup for HGNN and GNN baseline comparison

We benchmarked the models on datasets from above, which we split the training, validation and testing in a 3:1:1 ratio. We trained baseline models on a mixture of SNS, MNS, CNS, RCNS, and with NS ratio of 1:1. We evaluated the performance of the models with 5 binary classification metrics and tested with negative data generated from different NS. Additionally, we evaluate the model with whether the model collapses since some models tend to always predict one class. For the initial embedding of the nodes, we used the Morgan Fingerprint, ECFP6 (Rogers & Hahn, 2010), to ensure they have the same initial information before the propagation.

### A.5.3  Setup for benchmarking ChemHGNN

Same as the above setup. We evaluated the performance of the models with 5 binary classification metrics. Since NOCD outperforms other GNN baselines, to show the effectiveness of our ChemHGNN. To better understand the behavior of the model correlates the the training dataset, we investigate the factors related to the reaction type in the dataset. We classify the reaction by rxnfp to 1k template. In Fig. 11, we can see a clear pattern of imbalanced class distribution where the reaction template 672 dominates the data distribution in the dataset of 10k datapoints. We also observe several dominant reaction type like 586 and 274. Therefore, we construct new dataset from the top 3 frequent reaction template (RT 274, RT 586, RT 672) of 10k datapoints and investigate the performance.

### A.5.4  Setup for benchmarking Negative Sampling

Since RCNS involves the creation of virtual nodes, we created two hypergraph reaction network, one with the virtual nodes from RCNS, and the other without. We benchmarked the ChemHGNN trained on two different hypergraph reaction networks in a mixed NS with RCNS and without negative sampling, and tested on MNS, CNS and SNS.

### .1  Hyperparameters and Evaluation Details

### .1.1  Hyperparameter Settings

The hyperparameters for our models were tuned using grid search. The selected configurations are summarized in Tables 3 and 4.

### .1.2  Binary Conversion for Metrics Computation

To compute precision, recall, and F-measure, predictions (probabilities) are converted to binary labels using a threshold of 0.5:

Table 3: HGNN Hyperparameters

| Parameter | Value |
|---|---|
| Input size | 1024 |
| Output size | `args.dim_vertex` |
| Hidden dimensions | 16 |
| Dropout rate | 0.5 |
| Vertex embedding dimension | 1024 |
| GNN embedding dimension | 5 |
| Key-value dimension | 256 |
| Aggregation method | sum |
| Classification layers | [1024, 256, 16, 1] |
| Learning rate | 0.0001 |
| Batch size | 16 |
| Epochs | 50 |
| Lambda | 0.5 |

Table 4: WLN (pretrained) Hyperparameters

| Parameter | Value |
|---|---|
| Batch size | 20 |
| Hidden size | 300 |
| Max norm | 5.0 |
| Node input features | 82 |
| Edge input features | 6 |
| Node pair input features | 10 |
| Node output features | 300 |
| Number of layers | 3 |
| Number of tasks | 5 |
| Learning rate | 0.001 |
| Number of epochs | 18 |
| Decay every | 10000 |
| Learning rate decay factor | 0.9 |

- If the predicted probability $\geq 0.5$, the label is 1 (positive class).

- If the predicted probability $< 0.5$, the label is 0 (negative class).

This standard conversion allows accurate evaluation of classification metrics.

## .2 MORE EXPERIMENTAL RESULTS AND VISUALIZATION

### .2.1 HGNN AND GNN BASELINE COMPARISON

Table 5: HGNN and GNN baseline (HGNN Feng et al. (2019), NOCD Shchur & Günnemann (2019), GCN Kipf & Welling (2016), GAT Veličković et al. (2017)) performance comparison across different dataset scales and NS strategies. Models are trained in a mixture of NS strategy (MNS, CNS, SNS and RCNS). The **bold** numbers are the best result in each metric per NS strategy. We also include the "Model Collapse" metric to evaluate whether collapse occurs during testing.

| Dataset | NS Strategy | Model | Model Collapse | Recall | F1 | Specificity |
|---------|-------------|-------|----------------|--------|------|-------------|
| USPTO-1000 | MNS | HGNN | ✗ | 0.6392 | 0.6492 | 0.6701 |
| | | NOCD | ✗ | 0.0913 | 0.1639 | **0.9381** |
| | | GCN | ✓ | **1.0000** | **0.6667** | 0.0000 |
| | | GAT | ✗ | 0.2593 | 0.3805 | 0.7423 |
| | SNS | HGNN | ✗ | 0.6392 | 0.7750 | **0.9897** |
| | | NOCD | ✗ | 0.0913 | 0.1636 | 0.9433 |
| | | GCN | ✓ | **1.0000** | **0.6667** | 0.0000 |
| | | GAT | ✗ | 0.2593 | 0.3968 | 0.8814 |
| | CNS | HGNN | ✗ | 0.6392 | 0.5451 | 0.2938 |
| | | NOCD | ✗ | 0.0913 | 0.1574 | **0.8505** |
| | | GCN | ✓ | **1.0000** | **0.6667** | 0.0000 |
| | | GAT | ✗ | 0.2593 | 0.3671 | 0.6186 |
| USPTO-5000 | MNS | HGNN | ✓ | 0.9927 | 0.6638 | 0.0021 |
| | | NOCD | ✓ | 0.0021 | 0.0042 | **0.9979** |
| | | GCN | ✓ | **1.0000** | **0.6667** | 0.0000 |
| | | GAT | ✓ | 0.0361 | 0.0692 | 0.9822 |
| | SNS | HGNN | ✗ | **0.9927** | **0.8591** | 0.6817 |
| | | NOCD | ✓ | 0.0021 | 0.0042 | **0.9979** |
| | | GCN | ✓ | 1.0000 | 0.6667 | 0.0000 |
| | | GAT | ✓ | 0.0361 | 0.0696 | 0.9958 |
| | CNS | HGNN | ✓ | 0.9926 | 0.6655 | 0.0094 |
| | | NOCD | ✓ | 0.0021 | 0.0042 | **0.9937** |
| | | GCN | ✓ | **1.0000** | **0.6667** | 0.0000 |
| | | GAT | ✓ | 0.0361 | 0.0686 | 0.9581 |
| USPTO-10000 | MNS | HGNN | ✗ | 0.0938 | 0.1681 | 0.9775 |
| | | NOCD | ✓ | 0.0021 | 0.0042 | **1.0000** |
| | | GCN | ✓ | **1.0000** | **0.6667** | 0.0000 |
| | | GAT | ✓ | 0.0059 | 0.0117 | 0.9979 |
| | SNS | HGNN | ✗ | 0.0938 | 0.1371 | 0.7253 |
| | | NOCD | ✓ | 0.0021 | 0.0042 | **1.0000** |
| | | GCN | ✓ | **1.0000** | **0.6667** | 0.0000 |
| | | GAT | ✓ | 0.0059 | 0.0116 | 0.9916 |
| | CNS | HGNN | ✗ | 0.0938 | 0.1600 | 0.9214 |
| | | NOCD | ✓ | 0.0021 | 0.0042 | **0.9984** |
| | | GCN | ✓ | **1.0000** | **0.6667** | 0.0000 |
| | | GAT | ✓ | 0.0059 | 0.0117 | 1.0000 |

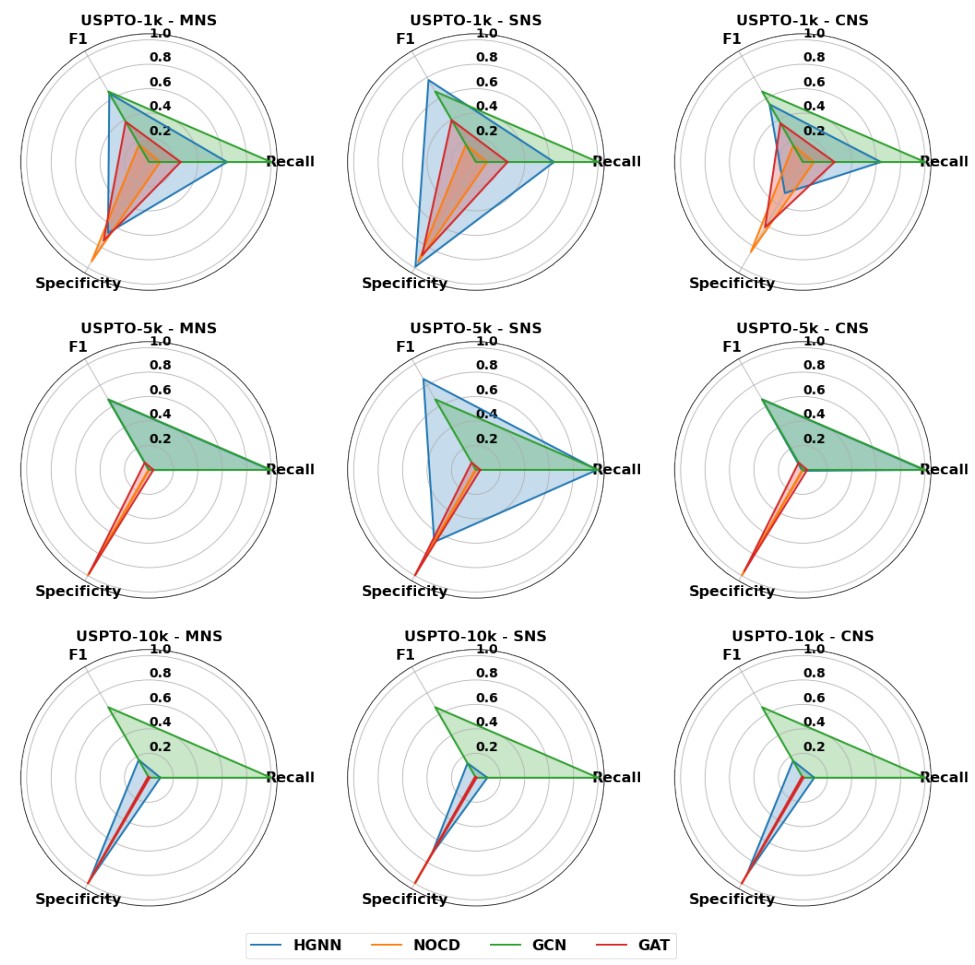

Figure 12: Radar plot of baseline model comparisons across different data scales and NS strategies. The extremely thin shaded regions for certain models (e.g., NOCD, GAT) highlight sharp metric imbalance, perfect specificity with near-zero recall, and green shaded area (e.g., GCN) highlights high recall, near-zero specificity region, both indicating model collapse.

.2.2 CHEMHGNN COMPARISON ACROSS DIFFERENT DATA SCALES

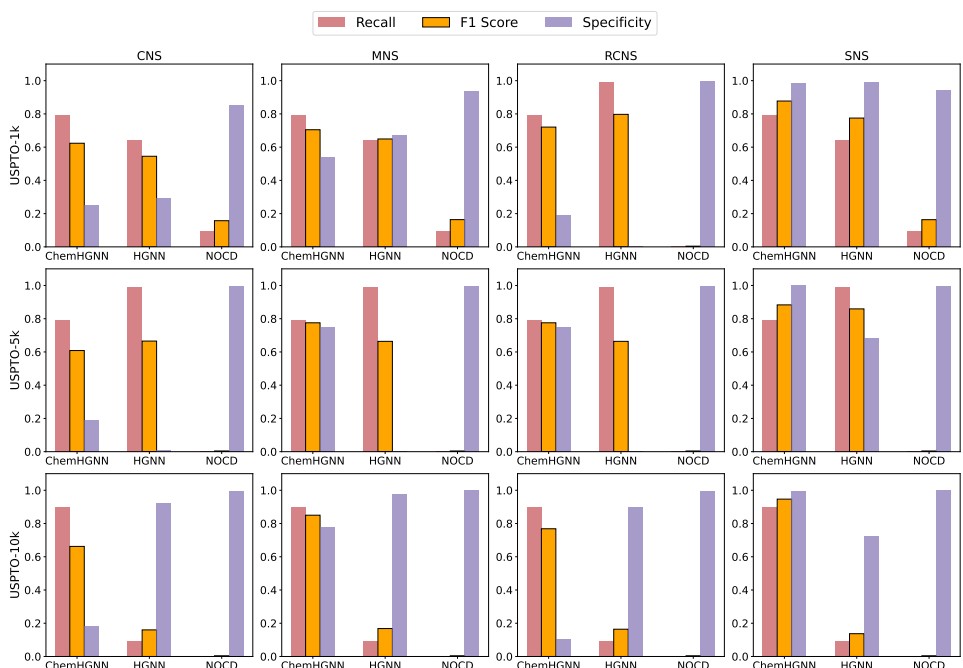

Figure 13: Bar plot of ChemHGNN comparisons across different data scales and NS strategies. Due to model collapse, some of the bars are **invisible** (e.g., in NOCD). Among the plotted evaluation metrics, the F1 Score is **highlighted** using a distinct yellow bar with a black border to draw attention to its importance, particularly relevant in evaluating model performance under class imbalance or skewed data.

Table 6: Models Trained on USPTO-1000 and tested with mixed NS strategies. **Bold** numbers indicate the best results per metric within each strategy. It includes mean ± standard deviation across 5 splits.

| NS Strat | Model | Accuracy | Precision | Recall | F1 Score | Specificity |
|---|---|---|---|---|---|---|
| MNS | ChemHGNN | **0.6675** ± 0.0453 | 0.6337 ± 0.0431 | **0.7938** ± 0.0574 | **0.7048** ± 0.0279 | 0.5412 ± 0.1176 |
| | HGNN | 0.6546 ± 0.0104 | 0.6569 ± 0.0060 | 0.6392± 0.0252 | 6492 ± 0.0108 | 0.6701 ± 0.0157 |
| | NOCD | 0.3358 ± 0.0576 | **0.8000** ± 0.0223 | 0.0913 ± 0.0140 | 0.1639 ± 0.0201 | **0.9381** ± 0.0261 |
| SNS | ChemHGNN | **0.8892** ± 0.0218 | **0.9809** ± 0.0239 | **0.7938** ± 0.0574 | **0.8775** ± 0.0260 | 0.9845 ± 0.0245 |
| | HGNN | 0.8144 ± 0.0218 | 0.9841 ± 0.0483 | 0.6392 ± 0.0252 | 0.7750 ± 0.0183 | **0.9897** ± 0.0686 |
| | NOCD | 0.3343 ± 0.0572 | 0.7857 ± 0.0211 | 0.0913 ± 0.0140 | 0.1636 ± 0.0203 | 0.9433 ± 0.0252 |
| CNS | ChemHGNN | **0.5206** ± 0.0086 | 0.5133 ± 0.0502 | **0.7938** ± 0.0574 | **0.6235** ± 0.0220 | 0.2474 ± 0.0601 |
| | HGNN | 0.4665 ± 0.0086 | 0.4751 ± 0.0502 | 0.6392 ± 0.0252 | 0.5451 ± 0.1819 | 0.2938 ± 0.3694 |
| | NOCD | 0.3033 ± 0.0500 | **0.5714** ± 0.0231 | 0.0913 ± 0.0140 | 0.1574 ± 0.0194 | **0.8505** ± 0.0268 |
| RCNS | ChemHGNN | **0.5739** ± 0.0120 | 0.6553 ± 0.0068 | **0.7938** ± 0.0574 | **0.7179** ± 0.0105 | 0.1000 ± 0.0209 |
| | HGNN | 0.5106 ± 0.0120 | 0.6425 ± 0.0068 | 0.6392 ± 0.0252 | 0.6408 ± 0.0105 | 0.2333 ± 0.0209 |
| | NOCD | 0.2273 ± 0.0478 | **0.9167** ± 0.0161 | 0.0913 ± 0.0140 | 0.1660 ± 0.0203 | **0.9333** ± 0.0251 |

Table 7: Models Trained on USPTO-5000 and tested with mixed NS strategies. **Bold** numbers indicate the best result per metric in each strategy. It includes include mean ± standard deviation across 5 splits.

| NS Strat | Model | Accuracy | Precision | Recall | F1 Score | Specificity |
|---|---|---|---|---|---|---|
| MNS | ChemHGNN | **0.7707** ± 0.0325 | **0.7598** ± 0.0419 | **0.7916** ± 0.0132 | **0.7754** ± 0.0234 | 0.7497 ± 0.0689 |
| | HGNN | 0.5679 ± 0.0450 | 0.5971 ± 0.0899 | 0.7091 ± 0.3287 | 0.5726 ± 0.1703 | 0.4252 ± 0.3148 |
| | NOCD | 0.2858 ± 0.0536 | 0.4545 ± 0.0323 | 0.0021 ± 0.0013 | 0.0042 ± 0.0030 | **0.9979** ± 0.0012 |
| SNS | ChemHGNN | **0.8953** ± 0.0061 | **0.9987** ± 0.0058 | 0.7916 ± 0.0132 | **0.8832** ± 0.0070 | **0.9990** ± 0.0052 |
| | HGNN | 0.7260 ± 0.1539 | 0.6786 ± 0.1986 | 0.7091 ± 0.3287 | 0.6761 ± 0.2641 | 0.7422 ± 0.0759 |
| | NOCD | 0.2867 ± 0.0538 | 0.6250 ± 0.0319 | 0.0021 ± 0.0013 | 0.0042 ± 0.0029 | 0.9979 ± 0.0011 |
| CNS | ChemHGNN | 0.4906 ± 0.0115 | 0.4941 ± 0.0068 | **0.7916** ± 0.0132 | **0.6085** ± 0.0073 | 0.1895 ± 0.0229 |
| | HGNN | 0.5084 ± 0.0086 | 0.5272 ± 0.0502 | 0.7091 ± 0.3287 | 0.5412 ± 0.1819 | 0.3087 ± 0.3694 |
| | NOCD | 0.2867 ± 0.0535 | **0.6250** ± 0.0318 | 0.0021 ± 0.0013 | 0.0042 ± 0.0029 | **0.9937** ± 0.0013 |
| RCNS | ChemHGNN | 0.5905 ± 0.0120 | 0.6614 ± 0.0068 | **0.7916** ± 0.0132 | **0.7207** ± 0.0105 | 0.1870 ± 0.0209 |
| | HGNN | 0.6075 ± 0.1187 | 0.6710 ± 0.0164 | 0.7091 ± 0.3287 | 0.6337 ± 0.2603 | 0.2847 ± 0.3342 |
| | NOCD | 0.1672 ± 0.0561 | 0.6250 ± 0.0321 | 0.0021 ± 0.0013 | 0.0042 ± 0.0030 | **0.9958** ± 0.0010 |

Table 8: Models Trained on USPTO-10000 and tested with mixed NS strategies. **Bold** numbers indicate the best result per metric per NS strategy. It includes include mean ± standard deviation across 5 splits.

| NS Strat | Model | Accuracy | Precision | Recall | F1 Score | Specificity |
|---|---|---|---|---|---|---|
| MNS | ChemHGNN | **0.8412** ± 0.0274 | 0.8051 ± 0.0460 | **0.9004** ± 0.0250 | **0.8501** ± 0.0234 | 0.7820 ± 0.0671 |
| | HGNN | 0.5305 ± 0.0184 | 0.7772 ± 0.1763 | 0.2729 ± 0.3940 | 0.2672 ± 0.2201 | 0.7885 ± 0.4825 |
| | NOCD | 0.2870 ± 0.0531 | **0.8333** ± 0.0293 | 0.0021 ± 0.0015 | 0.0042 ± 0.0028 | **1.0000** ± 0.0006 |
| SNS | ChemHGNN | **0.9494** ± 0.0123 | **0.9983** ± 0.0007 | **0.9004** ± 0.0250 | **0.9468** ± 0.0139 | 0.9984 ± 0.0007 |
| | HGNN | 0.5048 ± 0.1748 | 0.3737 ± 0.2088 | 0.2713 ± 0.3940 | 0.2983 ± 0.3023 | 0.7388 ± 0.0356 |
| | NOCD | 0.2870 ± 0.0532 | 0.8333 ± 0.0301 | 0.0021 ± 0.0014 | 0.0042 ± 0.0027 | **1.0000** ± 0.0005 |
| CNS | ChemHGNN | **0.5409** ± 0.0115 | 0.5238 ± 0.0068 | **0.9004** ± 0.0250 | **0.6623** ± 0.0073 | 0.1813 ± 0.0264 |
| | HGNN | 0.5116 ± 0.0055 | 0.5655 ± 0.0393 | 0.2729 ± 0.3940 | 0.2625 ± 0.2229 | 0.7506 ± 0.3844 |
| | NOCD | 0.2871 ± 0.0532 | **0.9091** ± 0.0310 | 0.0021 ± 0.0014 | 0.0042 ± 0.0027 | **0.9984** ± 0.0009 |
| RCNS | ChemHGNN | **0.6370** ± 0.0120 | 0.6703 ± 0.0068 | **0.9004** ± 0.0250 | **0.7685** ± 0.0105 | 0.1039 ± 0.0209 |
| | HGNN | 0.3614 ± 0.0029 | 0.6667 ± 0.0177 | 0.2729 ± 0.3940 | 0.2911 ± 0.2759 | 0.9247 ± 0.0369 |
| | NOCD | 0.1665 ± 0.0531 | **0.8333** ± 0.0292 | 0.0021 ± 0.0014 | 0.0042 ± 0.0028 | **0.9979** ± 0.0007 |

### .2.3 CHEMHGNN COMPARISON ACROSS DIFFERENT TESTING NS STRATEGIES

Table 9: Models Trained with a mixture of NS strategies(MNS, SNS, CNS and RCNS) test with MNS. **Bold** number are the best result in each metric per NS Strategy

| Dataset | Model | Accuracy | Precision | Recall | F1 Score | Specificity |
|---|---|---|---|---|---|---|
| USPTO-1k | ChemHGNN | **0.6675** | 0.6337 | **0.7938** | **0.7048** | 0.5412 |
| | HGNN | 0.6546 | 0.6596 | 0.6392 | 0.6492 | 0.6701 |
| | NOCD | 0.3358 | **0.8000** | 0.0913 | 0.1639 | **0.9381** |
| USPTO-5k | ChemHGNN | **0.7707** | **0.7598** | 0.7916 | **0.7754** | 0.7497 |
| | HGNN | 0.4974 | 0.4987 | **0.9927** | 0.6638 | 0.0021 |
| | NOCD | 0.2858 | 0.4545 | 0.0021 | 0.0042 | **0.9979** |
| USPTO-10k | ChemHGNN | **0.8412** | 0.8051 | **0.9004** | **0.8501** | 0.7820 |
| | HGNN | 0.5356 | 0.8063 | 0.0938 | 0.1681 | 0.9775 |
| | NOCD | 0.2870 | **0.8333** | 0.0021 | 0.0042 | **1.0000** |

Table 10: Models Trained with a mixture of NS strategies(MNS, SNS, CNS and RCNS) test with SNS. **Bold** number are the best result in each metric per NS Strategy

| Dataset | Model | Accuracy | Precision | Recall | F1 Score | Specificity |
|---|---|---|---|---|---|---|
| USPTO-1k | ChemHGNN | **0.8892** | 0.9809 | **0.7938** | **0.8775** | 0.9845 |
| | HGNN | 0.8144 | **0.9841** | 0.6392 | 0.7750 | **0.9897** |
| | NOCD | 0.3343 | 0.7857 | 0.0913 | 0.1636 | 0.9433 |
| USPTO-5k | ChemHGNN | **0.8953** | **0.9987** | 0.7916 | **0.8832** | **0.9990** |
| | HGNN | 0.8371 | 0.7572 | **0.9927** | 0.8591 | 0.6817 |
| | NOCD | 0.2867 | 0.6250 | 0.0021 | 0.0042 | 0.9979 |
| USPTO-10k | ChemHGNN | **0.9494** | **0.9983** | **0.9004** | **0.9468** | 0.9984 |
| | HGNN | 0.4096 | 0.2546 | 0.0938 | 0.1371 | 0.7253 |
| | NOCD | 0.2870 | 0.8333 | 0.0021 | 0.0042 | **1.0000** |

Table 11: Models Trained with a mixture of NS strategies(MNS, SNS, CNS and RCNS) test with CNS. **Bold** number are the best result in each metric per NS Strategy

| Dataset | Model | Accuracy | Precision | Recall | F1 Score | Specificity |
|---|---|---|---|---|---|---|
| USPTO-1k | ChemHGNN | **0.5206** | 0.5133 | **0.7938** | **0.6235** | 0.2474 |
| | HGNN | 0.4665 | 0.4751 | 0.6392 | 0.5451 | 0.2938 |
| | NOCD | 0.3033 | **0.5714** | 0.0913 | 0.1574 | **0.8505** |
| USPTO-5k | ChemHGNN | 0.4906 | 0.4941 | 0.7916 | 0.6085 | 0.1895 |
| | HGNN | **0.5010** | 0.5005 | **0.9926** | **0.6655** | 0.0094 |
| | NOCD | 0.2867 | **0.6250** | 0.0021 | 0.0042 | **0.9937** |
| USPTO-10k | ChemHGNN | **0.5409** | 0.5238 | **0.9004** | **0.6623** | 0.1813 |
| | HGNN | 0.5076 | 0.5440 | 0.0938 | 0.1600 | 0.9214 |
| | NOCD | 0.2871 | **0.9091** | 0.0021 | 0.0042 | **0.9984** |

Table 12: Models Trained with a mixture of NS strategies(MNS, SNS, CNS and RCNS) test with RCNS. **Bold** number are the best result in each metric per NS Strategy

| Dataset | Model | Accuracy | Precision | Recall | F1 Score | Specificity |
|---|---|---|---|---|---|---|
| USPTO-1k | ChemHGNN | **0.5905** | **0.6614** | **0.7916** | **0.7207** | 0.1870 |
| | HGNN | 0.5106 | 0.6425 | 0.6392 | 0.6408 | 0.2333 |
| | NOCD | 0.1672 | 0.6250 | 0.0021 | 0.0042 | **0.9958** |
| USPTO-5k | ChemHGNN | **0.7707** | **0.7598** | 0.7916 | **0.7754** | 0.7497 |
| | HGNN | 0.4974 | 0.4987 | **0.9927** | 0.6638 | 0.0021 |
| | NOCD | 0.2858 | 0.4545 | 0.0021 | 0.0042 | **0.9979** |
| USPTO-10k | ChemHGNN | **0.6370** | 0.6703 | **0.9004** | **0.7685** | 0.1039 |
| | HGNN | 0.3602 | 0.6533 | 0.0938 | 0.1641 | 0.8993 |
| | NOCD | 0.1665 | **0.8333** | 0.0021 | 0.0042 | **0.9979** |

### .2.4 CHEMHGNN AND GNN COMPARISON ALONG REACTION TEMPLATE

Table 13: Model trained on 10k datapoints of different reaction template, and tested on different NS strategies. **Bold** number are the best result in each metric per dataset.

| Dataset | Model | NS Strat | Accuracy | Precision | Recall | F1 Score | Specificity | HGNN better? |
|---|---|---|---|---|---|---|---|---|
| RT 274 | ChemHGNN | MNS | 0.6579 | 0.6212 | 0.8092 | 0.7029 | 0.5066 | |
| | | SNS | **0.8783** | **0.9389** | **0.8092** | **0.8693** | **0.9474** | |
| | | CNS | 0.4868 | 0.4920 | 0.8092 | 0.6119 | 0.1645 | |
| | | RCNS | 0.5935 | 0.6340 | 0.8092 | 0.7110 | 0.2447 | ✓ |
| | NOCD | MNS | 0.5682 | 0.7327 | 0.6197 | 0.6715 | 0.4408 | |
| | | SNS | 0.6250 | 0.8090 | 0.6197 | 0.7018 | 0.6382 | |
| | | CNS | 0.5170 | 0.6754 | 0.6197 | 0.6463 | 0.2632 | |
| | | RCNS | 0.5681 | 0.7952 | 0.6197 | 0.6966 | 0.3617 | |
| RT 586 | ChemHGNN | MNS | 0.8723 | 0.8507 | 0.9030 | 0.8761 | 0.8416 | |
| | | SNS | **0.9486** | **0.9936** | **0.9030** | **0.9462** | **0.9942** | |
| | | CNS | 0.5737 | 0.5444 | 0.9030 | 0.6793 | 0.2443 | |
| | | RCNS | 0.6126 | 0.6376 | 0.9030 | 0.7475 | 0.1079 | ✓ |
| | NOCD | MNS | 0.5036 | 0.7190 | 0.4987 | 0.5889 | 0.5157 | |
| | | SNS | 0.5505 | 0.7944 | 0.4987 | 0.6128 | 0.6792 | |
| | | CNS | 0.4729 | 0.6770 | 0.4987 | 0.5743 | 0.4088 | |
| | | RCNS | 0.4835 | 0.7912 | 0.4987 | 0.6118 | 0.4157 | |
| RT 672 | ChemHGNN | MNS | 0. 6118 | 0.5789 | 0.8199 | 0.6787 | 0.4037 | |
| | | SNS | **0.8882** | **0.9496** | **0.8199** | **0.8800** | **0.9565** | |
| | | CNS | 0.4875 | 0.4925 | 0.8199 | 0.6154 | 0.1553 | |
| | | RCNS | 0.6100 | 0.6701 | 0.8199 | 0.7374 | 0.1875 | ✓ |
| | NOCD | MNS | 0.5699 | 0.7445 | 0.6020 | 0.6657 | 0.4907 | |
| | | SNS | 0.6290 | 0.8299 | 0.6020 | 0.6978 | 0.6957 | |
| | | CNS | 0.5197 | 0.6848 | 0.6020 | 0.6408 | 0.3168 | |
| | | RCNS | 0.5597 | 0.8213 | 0.6020 | 0.6948 | 0.3500 | |

### .2.5 SORT OUT BLOCK ANALYSIS

Table 14: (a) Best score comparison of ChemHGNN models trained on different scales of dataset and across # of iterations to search new combinations. (b)Best score comparison of sort out function Simulated Annealing (SA) with random selection on ChemHGNN trained on USPTO-1k across different # of iterations.

| Model | # Iters | SA |
|---|---|---|
| ChemHGNN_1k | 1M | 0.577 |
| | 2M | 0.577 |
| | 3M | 0.576 |
| ChemHGNN_5k | 1M | 0.577 |
| | 2M | 0.580 |
| | 3M | 0.588 |
| ChemHGNN_10k | 1M | 0.582 |
| | 2M | 0.578 |
| | 3M | 0.582 |
| | 5M | 0.586 |

Table 15: *

**(a)** SA performance across models and iterations.

| Model | # Iters | SA | Random |
|---|---|---|---|
| ChemHGNN_1000 | 10k | 0.574 | 0.568 |
| | 1M | 0.576 | 0.574 |
| | 2M | 0.577 | 0.575 |

Table 16: *

**(b)** SA vs. Random on USPTO-1000.

.2.6 RCNS ANALYSIS AND ABLATION STUDIES

Table 17: ChemHGNN trained under different negative sampling (NS) strategies **with or without** reaction center-aware negative sampling (RCNS) across three dataset scales.The **bold** number represents the best result in each metric per dataset.

| Dataset | With RCNS | NS Strategy | Accuracy | Precision | Recall | F1 | Specificity | $\Delta\overline{F}_1$ |
|---|---|---|---|---|---|---|---|---|
| USPTO-1k | ✗ | MNS | 0.6650 | 0.6260 | 0.8200 | 0.7100 | 0.5100 | -0.55% |
| | | CNS | 0.5000 | 0.5000 | 0.8200 | 0.6212 | 0.1800 | |
| | | SNS | **0.8975** | 0.9704 | **0.8200** | **0.8889** | 0.9750 | |
| | ✓ | MNS | 0.6675 | 0.6337 | 0.7938 | 0.7048 | 0.5412 | |
| | | CNS | 0.5206 | 0.5133 | 0.7938 | 0.6235 | 0.2474 | |
| | | SNS | 0.8892 | **0.9809** | 0.7938 | 0.8775 | **0.9845** | |
| USPTO-5k | ✗ | MNS | 0.8000 | 0.7947 | 0.8090 | 0.8018 | 0.7910 | +0.58% |
| | | CNS | 0.5035 | 0.5022 | 0.8090 | 0.6197 | 0.1980 | |
| | | SNS | 0.9040 | **0.9988** | 0.8090 | 0.8939 | **0.9990** | |
| | ✓ | MNS | 0.7764 | 0.7454 | 0.8398 | 0.7898 | 0.7131 | |
| | | CNS | 0.5021 | 0.5013 | 0.8398 | 0.6278 | 0.1644 | |
| | | SNS | **0.9188** | 0.9975 | **0.8398** | **0.9119** | 0.9979 | |
| USPTO-10k | ✗ | MNS | 0.7262 | 0.6677 | 0.9010 | 0.7670 | 0.5515 | +7.75% |
| | | CNS | 0.5285 | 0.5163 | 0.9010 | 0.6565 | 0.1560 | |
| | | SNS | 0.9295 | 0.9555 | **0.9010** | 0.9274 | 0.9580 | |
| | ✓ | MNS | 0.8412 | 0.8051 | 0.9004 | 0.8501 | 0.7820 | |
| | | CNS | 0.5409 | 0.5238 | 0.9004 | 0.6623 | 0.1813 | |
| | | SNS | **0.9494** | **0.9983** | 0.9004 | **0.9468** | **0.9984** | |

Table 18: Ablation study on ChemHGNN trained on USPTO-10k and mix and tested with mix negative.

| Blocks | NS Strat | Accuracy | Precision | Recall | F1 Score | Specificity | decrement in F1 |
|---|---|---|---|---|---|---|---|
| +WLN +SUM +MSE | MNS | 0.8412 | 0.8051 | 0.9004 | 0.8501 | 0.7820 | – |
| | SNS | 0.9494 | 0.9983 | 0.9004 | 0.9468 | 0.9984 | – |
| | CNS | 0.5409 | 0.5238 | 0.9004 | 0.6623 | 0.1813 | – |
| | RCNS | 0.6370 | 0.6703 | 0.9004 | 0.7685 | 0.1039 | – |
| +WLN +SUM −MSE | MNS | 0.8048 | 0.7656 | 0.8784 | 0.8182 | 0.7311 | -3.75% |
| | SNS | 0.9374 | 0.9958 | 0.8784 | 0.9334 | 0.9963 | -1.41% |
| | CNS | 0.5081 | 0.5047 | 0.8784 | 0.6410 | 0.1378 | -3.21% |
| | RCNS | 0.6247 | 0.6667 | 0.8784 | 0.7580 | 0.1113 | -1.36% |
| +WLN -SUM -MSE | MNS | 0.6659 | 0.8112 | 0.4324 | 0.5641 | 0.8994 | -33.6% |
| | SNS | 0.7138 | 0.9892 | 0.4324 | 0.6018 | 0.9953 | -36.4% |
| | CNS | 0.4914 | 0.4902 | 0.4324 | 0.4595 | 0.5503 | -30.6% |
| | RCNS | 0.4788 | 0.6718 | 0.4324 | 0.5261 | 0.5726 | -31.5% |
| -WLN -SUM -MSE | MNS | 0.5356 | 0.8063 | 0.0938 | 0.1681 | 0.9775 | -80.2% |
| | SNS | 0.4096 | 0.2546 | 0.0938 | 0.1371 | 0.7254 | -85.5% |
| | CNS | 0.5076 | 0.5441 | 0.0938 | 0.1600 | 0.9214 | -75.8% |
| | RCNS | 0.3602 | 0.6533 | 0.0938 | 0.1641 | 0.8993 | -78.6% |

.3  MORE TSNE PLOTS

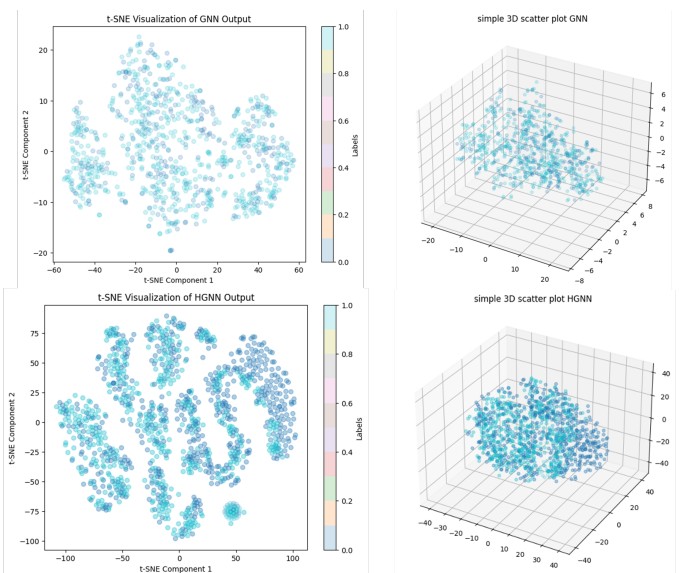

Figure 14: 2D, 3D TSNE comparison between reaction representation from baseline model of GNN (top2) and HGNN (bottom2) on USPTO-1000

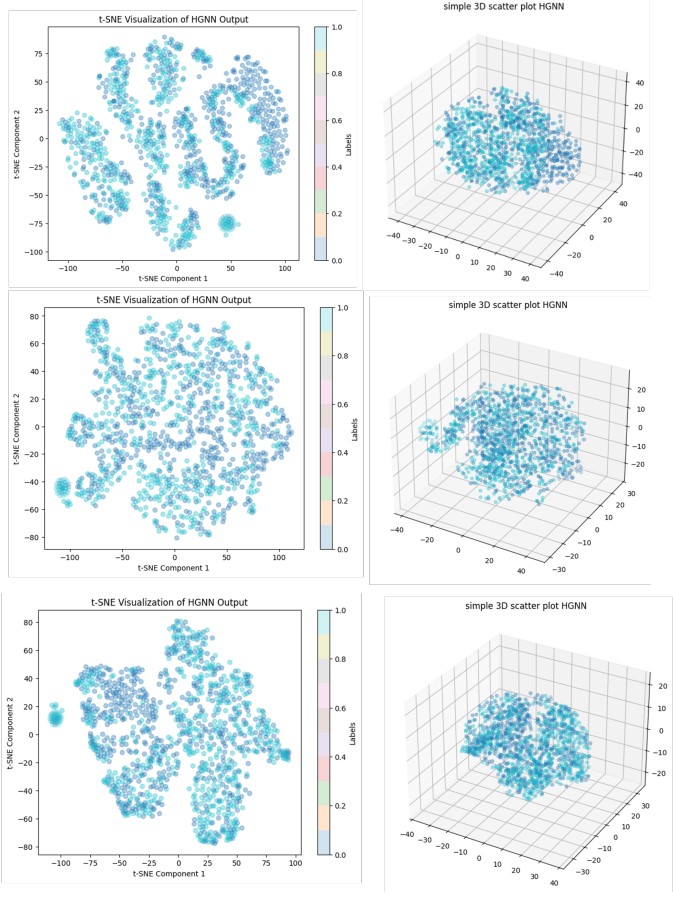

Figure 15: 2D, 3D TSNE comparison between reaction representation from baseline model of HGNN (top2) and ChemHGNN without MSE loss (mid2) and ChemHGNN (bottom2) on USPTO-1000

## .4 SORT OUT BLOCK

### .4.1 MORE ABOUT SIMULATED ANNEALING (SA)

The objective function $f(s)$ measures the quality of a solution $s$. The goal of the algorithm is to minimize or maximize this function. The choice of the objective function depends on the specific optimization problem being addressed.

The algorithm starts with an initial solution $s$, which can be generated randomly or using a heuristic method. This starting point serves as the first candidate solution and influences the initial exploration of the search space.

Temperature $T$ is a control parameter that governs the probability of accepting worse solutions. A higher temperature increases the likelihood of accepting suboptimal solutions, promoting extensive exploration. As the algorithm progresses, the temperature is gradually reduced to encourage convergence.

The cooling rate $\alpha$ determines how quickly the temperature decreases. Typically, $\alpha$ is a value between 0 and 1, where a slower cooling rate (closer to 1) results in a more exhaustive search, while a faster cooling rate (closer to 0) leads to quicker convergence.

At each temperature level, the algorithm performs $L$ iterations to explore the neighborhood of the current solution. A larger $L$ allows for a more thorough examination of the search space, increasing the likelihood of finding a better solution.

A neighboring solution $s'$ is generated by applying small, randomized changes to the current solution $s$. The quality of the new solution is evaluated using the objective function.

The new solution is evaluated with the change in objective Function $\Delta E$:
$$\Delta E = f(s') - f(s). \tag{14}$$

If $\Delta E < 0$, the new solution $s'$ is accepted since it improves the objective function. Otherwise, the algorithm uses a probabilistic criterion to decide whether to accept the worse solution. When $\Delta E \geq 0$, the new solution is accepted with a probability given by:
$$P = \exp\left(-\frac{\Delta E}{T}\right). \tag{15}$$

This probabilistic acceptance mechanism prevents the algorithm from getting trapped in local optima, allowing it to explore more diverse regions of the search space.

The algorithm terminates when a predefined stopping criterion is met. Common criteria include reaching a minimal temperature, exceeding a maximum number of iterations, or detecting convergence.

## .5 LIMITATIONS AND OUTLOOK

While this work presents several contributions, it is not without limitations. Key areas for improvement include scalability, the number of optimization iterations, and the definition of the chemical space explored.

### .5.1 SCALABILITY

As the reaction space expands, the memory footprint of key components—such as the original embedding $X$, the incidence matrix $H$, and matrices involved in the Laplacian computation—grows exponentially. Although sparse matrix techniques help mitigate this issue, they only provide limited relief. In our current setup, using a single NVIDIA A100 GPU with 80 GB of VRAM, we are constrained to exploring a chemical space of up to 20,000 reactions. Future work could benefit from designing more scalable architectures or leveraging distributed computing frameworks, which remain a viable option given the increasing availability of computational resources.

### .5.2 NUMBER OF ITERATIONS

The current optimization relies on Simulated Annealing (SA), a heuristic approach that is not guaranteed to be optimal for NP-complete problems. Although SA yields performance improvements,

it requires a large number of iterations to achieve high-quality selections. Future efforts may focus on developing more efficient and effective combination strategies, potentially informed by node representations and learning-guided exploration techniques.

### .5.3 DEFINITION OF CHEMICAL SPACE

The definition of chemical space plays a critical role in reaction discovery. An appropriately selected space can significantly increase the likelihood of uncovering novel reactions. However, the criteria for constructing an optimal chemical space remain an open question. Further investigation is needed to understand the characteristics that lead to more fruitful exploration.

### .6 COMPUTATIONAL RESOURCES

All experiments were conducted using a single NVIDIA A100 GPU with 80 GB of VRAM. The maximum runtime for any experiment was approximately 12 hours.

### .7 LLM USAGE DISCLOSURE

In accordance with ICLR 2026 policy, we disclose the use of Large Language Models (LLMs) during manuscript preparation. LLMs were employed only to aid in editing and polishing the writing. All conceptual contributions, methodological developments, experiments, and analyses were conducted by the authors.

