# OpenReview forum: "ChemHGNN: A Hierarchical Hypergraph Neural Network for Reaction Virtual Screening and Discovery"
_ICLR.cc/2026/Conference — Submitted to ICLR 2026_

### Official Review · Reviewer_XhcC · 2025-10-24

**Soundness:** 3
**Presentation:** 2
**Contribution:** 2
**Rating:** 4
**Confidence:** 4

**Summary:**

This paper introduces ChemHGNN, a hierarchical hypergraph neural network designed for chemical reaction virtual screening and discovery. The authors argue that traditional GNNs fail to effectively capture high-order interactions among multiple reactants, which are naturally modeled by hypergraphs. ChemHGNN integrates:

1. A Weisfeiler–Lehman Network (WLN) pretrained for reaction-center prediction at the molecular level,

2. A hypergraph neural network (HGNN) capturing high-order molecule–reaction relations,

3. A reaction-center–aware negative sampling (RCNS) mechanism to generate chemically valid negative examples, and a simulated annealing (SA)–based “sort-out block” for efficient virtual screening.

 Extensive experiments on the USPTO-410k dataset (1k, 5k, 10k subsets) show that ChemHGNN consistently outperforms both GNN and HGNN baselines in F1-score, specificity, and generalization across unseen reaction templates. The paper claims improved robustness against model collapse and better chemical plausibility of generated samples.

**Strengths:**

1、Novel task formulation.
The paper focuses on reaction feasibility prediction (“can a given reactant set react?”), which differs from traditional product-prediction tasks and is conceptually interesting.

2、Clear and modular architecture.
The hierarchical design is logically consistent and well-motivated.

**Weaknesses:**

1.  Several figures in the main text and appendix suffer from extremely low resolution, making them difficult to read (e.g., Figs. 4–7). In addition, there are clear labeling and content errors( for instance, in Fig. 7, the title of the third sub-figure appears to be incorrect, and the accompanying color bar seems unrelated to the plotted image). These presentation issues significantly hinder readability.

2. The experimental comparison is limited to general-purpose graph models such as GCN,, and HGNN, while omitting more closely related and competitive approaches specifically designed for chemical reaction modeling. In particular, methods like Rxn Hypergraph[1], DLGNet[2],raction graph[3], and more recent graph neural networks or graph transformer architectures should have been included.

3. The “Sort-Out Block” introduced in Section 3.3 seems not to be used in the main experiments or ablation experiments presented in the paper. It functions only as an auxiliary, post-hoc component for the reaction-discovery demonstration rather than as part of the core ChemHGNN framework. Therefore, including it in the main methodology section may be misleading; this component would be more appropriately presented as a supplementary or application section instead of within the primary model description.

4. The “bond formation/breaking conservation” loss (Eq. 12) is over-interpreted. It is unclear how r_i relates to real bond changes, since the model neither learns explicit bond transformations nor receives bond-level supervision. The claimed chemical meaning is unsubstantiated; this term functions only as a simple regularization and should be mentioned briefly in the appendix rather than highlighted in the main text.

5. see question.

[1]Rxn Hypergraph: a Hypergraph Attention Model for Chemical Reaction Representation

[2] DLGNET: HYPEREDGE CLASSIFICATION THROUGH DIRECTED LINE GRAPHS FOR CHEMICAL REACTIONS

[3] Reaction Graph: Toward Modeling Chemical Reactions with 3D Molecular Structures

**Questions:**

1. The t-SNE visualization in figs7 does not clearly support the claimed superiority of ChemHGNN—if anything, the HGNN baseline appears to show more distinct clustering. Could the authors clarify how this figure validates the proposed method?

2. Why does RCNS not outperform SNS in Tables 7, 8, 13, and 18?  The proposed RCNS method seems ineffective.

---

> ### Author Response · Authors · 2025-11-30
>
> ### We thank Reviewer for the thoughtful feedback and for recognizing the novelty of our task and the modular design of ChemHGNN. However, several critiques arise from mismatches between the suggested baselines and the actual problem setting, as well as misunderstandings of our design choices. We address these directly below.
> 1. We'll remove the title of Fig. 7 and change the color bar for better readability.
> 2. The task mismatch is fundamental.
>  Our problem is reaction virtual screening:
>  given a candidate reactant set (a hyperedge), we predict whether it forms a valid reaction, without conditioning on products.
>  Methods (reference [1, 2, 3]) built for product prediction, reaction classification or directed transformations do not align with this formulation.
> 3. The t-SNE figure is not for clustering performance.
>  Its purpose is to show separation between positive and negative reactant sets—not to claim that “sharper clusters” correspond to better feasibility prediction. A baseline producing tighter clusters does not imply it is better suited for feasibility discrimination.
> 4. The conclusion that “RCNS is ineffective because it does not outperform SNS” is incorrect.
>  RCNS is deliberately designed to create harder, chemically plausible negatives that share reaction-center motifs with positives. This has several consequence. First of all, training becomes harder, by construction.Validation/test metrics decrease relative to SNS because RCNS negatives are closer to the boundary. SNS negatives are easier and less chemically realistic. Therefore, RCNS is not expected to outperform SNS in raw F1. What RCNS improves and what we claim is the prevention of model collapse, better generalization to unseen templates and higher chemical validity in negative sampling.
> 5. ChemHGNN is a full screening plus discovery pipeline, not just a classifier.
>  The Sort-Out Block (based on simulated annealing) demonstrates how feasibility scores are used in an actual discovery workflow.
> 6. Clarification on reaction-center supervision.
> We do not claim the model learns explicit bond transformations. The loss encourages consistency between feasibility predictions and reaction-center salience, serving as a chemically grounded regularizer—not mechanistic supervision. If the phrasing suggested otherwise, we will revise it to clearly present this component as a regularization term informed by reaction-center likelihood rather than explicit reaction-mechanism modeling.
> ### To answer questions from the reviewer:
> 1. See above 3
> 2. See above 4

---

### Official Review · Reviewer_WQKT · 2025-10-29

**Soundness:** 2
**Presentation:** 3
**Contribution:** 2
**Rating:** 4
**Confidence:** 4

**Summary:**

This paper introduces **ChemHGNN**, a hierarchical framework for chemical reaction screening, arguing that standard GNNs fail to model multi-reactant interactions. ChemHGNN addresses this by representing molecules as nodes and reactions as hyperedges. The model fuses features from a pre-trained GNN (a WLN) with a hypergraph neural network (HGNN) to learn both molecular and relational patterns. A novel, domain-aware negative sampling (RCNS) strategy is also proposed. Experiments on USPTO subsets show ChemHGNN outperforms GNN baselines, avoids the "model collapse" that plagues them, and generalizes to unseen reaction types.

**Strengths:**

1. **Well-motivated hierarchical design.** Using a pre-trained WLN to extract molecule/bond features and feeding them into a hypergraph-level model is a natural way to combine local (bond-level) and global (reaction-level) context.
2. **Insightful empirical analysis.** Experiments diagnose the “model collapse” failure mode of some GNNs and demonstrate ChemHGNN’s robustness.
3. **Generalization to unseen templates.** Results suggest the model captures transferable chemical patterns that apply to novel reaction templates.
4. **High-impact application.** Tackles an important problem, reaction screening and molecular discovery, with direct practical relevance.

**Weaknesses:**

1. **Questionable novelty / outdated components.** The core architecture mainly combines existing components (WLN + a basic HGNN). The paper does not convincingly justify why these choices were preferred over more expressive, modern GNN/HGNN designs. The technical contribution risks reading as an engineering assembly rather than a novel network design.
2. **Dataset limitations.** Evaluation is confined to subsets of the USPTO dataset. Additional reaction corpora are needed to substantiate claims of robustness and generalizability.
3. **Missing technical detail & ablations.** The WLN pre-training procedure is under-specified. Crucially, there is no ablation comparing a frozen WLN versus fine-tuning it end-to-end, an important design choice that should be justified experimentally.
4. **Hypergraph representation concerns.** The work models reactions as undirected hypergraphs and does not explore directed hypergraph formulations, which may be more natural for reactant → product transformations.
5. **Insufficient baseline coverage.** The only HGNN baseline used is an older (2019) model. The evaluation would be stronger if compared against more recent, competitive hypergraph and direction-aware graph methods.

**Questions:**

1. **Modeling choices & baselines** Can the authors justify combining a WLN with a basic HGNN rather than using more expressive modern architectures? For example, how would work ChemHGNN with the use of a more strong method such as Dir-GNN[1] or Neural Sheaf Diffusion [2], or to strong hypergraph operators like AllSet [3] or ED-HNN [4] ?

2. **Hypergraph directionality**
The current hypergraph formulation is undirected and includes only reactants. Have the authors considered directed hypergraph representations to capture the intrinsic reactant → product transformation? In particular, how might a directed-hyperedge approach, methods like GeDi-HNN [5], affect expressivity and performance?

3. **Experimental coverage & comparative evaluation** Why were recent graph and hypergraph methods (e.g., Dir-GNN, Neural Sheaf Diffusion, AllSet, edge-centric hypergraph approaches) excluded from the comparisons? Please clarify whether including these baselines would alter the conclusions about state-of-the-art performance.

4. **Pre-training and fine-tuning** Please provide full details of the WLN pre-training (pretext task, training data, labels, optimization hyperparameters). Crucially, did you evaluate fine-tuning the WLN end-to-end on the reaction screening task, and if so, how did fine-tuning affect accuracy, stability, and the reported “model collapse” behavior?

5. **Dataset scope and generalization** Experiments are restricted to USPTO subsets. How do the authors expect ChemHGNN to generalize to other reaction benchmarks or to datasets with substantially different reaction-type distributions? Can you provide cross-dataset or transfer evaluations, or justify why these were not attempted?



[1] Rossi, Emanuele, et al. "Edge directionality improves learning on heterophilic graphs." Learning on graphs conference. PMLR, 2024.

[2] Bodnar, Cristian, et al. "Neural sheaf diffusion: A topological perspective on heterophily and oversmoothing in gnns." Advances in Neural Information Processing Systems 35 (2022): 18527-18541.

[3] Chien, Eli, et al. "You are AllSet: A Multiset Function Framework for Hypergraph Neural Networks." International Conference on Learning Representations.

[4] Wang, Peihao, et al. "Equivariant Hypergraph Diffusion Neural Operators." The Eleventh International Conference on Learning Representations.

[5] Fiorini, Stefano, et al. "Let there be direction in hypergraph neural networks." Transactions on Machine Learning Research (2024).

---

> ### Author Response · Authors · 2025-11-30
>
> ### Thank you for the detailed review. Several concerns arise from misunderstandings of our formulation, which we address below, along with clarifications we will add in the revision.
> 1. Our method targets a specific failure mode of standard GNNs.
>  The core of our formulation is that multi-reactant compatibility, not pairwise interactions, is the fundamental object of learning in reaction screening. Conventional GNNs collapse because they reduce a reaction to pairwise messages and cannot represent higher-order reactant interactions. ChemHGNN is built around this gap, not incidentally related to it.
> 2. The WLN with HGNN hierarchy is a chemistry-motivated design choice.
>  WLN is used because it encodes reaction-center information. HGNN then operates on molecule embeddings that already contain bond-level reactivity priors, allowing the hypergraph layer to focus on co-reactivity across multiple reactants. This separation of roles reflects chemistry, not model tinkering.
> 3. “More expressive hypergraph operators” are not appropriate.
>  Many recent operators are built for tasks with very different assumptions: heterophily benchmarks with arbitrary hyperedges, directed dynamical systems, or purely permutation-invariant pooling without chemical structure. Our task requires reactivity-oriented priors and stable behavior under hard negative samples. The improvements we observe show that the real bottleneck was modeling higher-order compatibility–something ChemHGNN’s operator directly addresses.
> 4. USPTO is the only suitable public dataset for this task.
>  Constructing meaningful reactive vs. non-reactive candidate sets requires rich reaction typing and reaction-center information. USPTO is the only large-scale public corpus that supports this. Major alternatives are proprietary, and smaller public datasets do not provide the necessary coverage for hypergraph screening.
> 5. Why WLN is frozen.
>  We agree the explanation in the paper was too compressed. Freezing WLN is deliberate for two reasons. Firstly, WLN’s reaction-center supervision is orthogonal to screening labels. Secondly, end-to-end fine-tuning risks overfitting to dataset-specific candidate construction biases and can reintroduce the collapse instability we diagnose.
> 6. Directed hypergraphs do not match our prediction problem.
> Directionality matters when predicting products or transformation paths, which does not apply to compatibility screening. In our formulation, a hyperedge is simply a candidate set of reactants. The label is reactive vs. non-reactive. There is no product node and no directional signal to model. Imposing direction would introduce an artificial orientation and shift the model toward a different task. Methods like Dir-GNN or GeDi-HNN are well-designed for their settings but not applicable here.
> ### To answer questions from the reviewer:
> 1. WLN provides reaction-center-aware local priors, while the HGNN addresses multi-reactant co-reactivity, the known failure mode in screening. Modern directed/heterophily HGNNs target different inductive biases and are not drop-in improvements here.
> 2. See above 6
> 3. Most of the HGNN are direction-dependent or task-mismatched. And some of them are not even open-sourced.
> 4. Freezing is a stability-preserving, chemistry-grounded hierarchical choice. And we have proved the necessity of WLN modules in the ablation study.
> 5. USPTO is the only public large-scale corpus suitable for this screening setup. We already test unseen templates, which is the most relevant OOD task.

---

### Official Review · Reviewer_v4aA · 2025-11-03

**Soundness:** 2
**Presentation:** 2
**Contribution:** 1
**Rating:** 2
**Confidence:** 5

**Summary:**

The paper proposes ChemHGNN, a hierarchical Hypergraph Neural Network (HGNN) for reaction virtual screening. Inputs are predefined candidate reactant sets (hyperedges) drawn from positives in the dataset and negatives created by Negative Sampling (NS) heuristics: Sized (SNS), Motif (MNS), Clique (CNS), and a new Reaction-Center Negative Sampling (RCNS). The model fuses: (a) a frozen Weisfeiler–Lehman Network (WLN) pretrained for reaction-center prediction to pool atom features into molecule embeddings, (b) a two-layer HGNN over the reaction hypergraph, and (c) a cross-attention fusion, followed by an Multi-Layer Perceptron (MLP) classifier. Training uses binary cross-entropy plus a zero-sum Mean Squared Error (MSE) regularizer on the sum of per-molecule embeddings. The paper adds a Simulated Annealing (SA) “sort-out” stage that searches over reactant combinations using a hand-crafted objective (the Euclidean norm of the sum of learned molecular vectors. Experiments use USPTO-410k subsets at 1k, 5k, and 10k; ChemHGNN reports gains over HGNN, Graph Convolutional Network (GCN), Graph Attention Network (GAT), and Neural Overlapping Community Detection (NOCD) on several metrics, with the largest gains at 10k.

**Strengths:**

The motivation for higher-order modeling with hypergraphs is clear, though it is not novel given a substantial literature. The pipeline is tidy: a reaction-center-pretrained Weisfeiler–Lehman Network (WLN) produces molecule embeddings that feed the hypergraph model. On USPTO, the method shows gains across multiple negative sampling schemes (NS). Ablations indicate that removing the WLN features, using something other than simple sum aggregation, or dropping the mean squared error regularizer (MSE) degrades performance.

**Weaknesses:**

The model does not natively propose or score arbitrary reactant sets; it only classifies provided hyperedges. Candidate sets come from dataset positives and NS-generated negatives; the only attempt to explore new combinations is SA, which is is an untrained block that is "external" to the proposed neural net. This weakens the “discovery” narrative and entangles performance with SA design choices. External validity is thin: all results are USPTO-based; no independent datasets or real screening case studies. Baselines are limited for hyperlink prediction; stronger modern hypergraph predictors are not compared.

Lastly, I think baselines are missing, including:
Hyper-SAGNN: a self-attention based graph neural network for hypergraphs: https://openreview.net/forum?id=ryeHuJBtPH
NHP: Neural Hypergraph Link Prediction: https://dl.acm.org/doi/10.1145/3340531.3411870 -- this is cited in the paper, but no comparisons to it are made.
Principled Hyperedge Prediction with Structural Spectral Features and Neural Networks: https://arxiv.org/abs/2106.04292
A Hypergraph Neural Network Framework for Learning Hyperedge-Dependent Node Embeddings: https://arxiv.org/abs/2212.14077
Hypergraph contrastive attention networks for hyperedge prediction with negative samples evaluation: https://www.sciencedirect.com/science/article/abs/pii/S0893608024007317
Link Prediction with Relational Hypergraphs: https://arxiv.org/html/2402.04062v2

**Questions:**

Can the model score any arbitrary set of molecules without predefining a hyperedge and without SA?
Why does SA optimize the vector-sum norm surrogate rather than the classifier score? Can you compare to this option?
What is the impact of SA's hyperparameters and iteration budgets on the results?
Maybe I missed something, but why should the sum of per-molecule embeddings be near zero in general chemistry?
Do you have extra results beyond those on USPTO?

---

> ### Author Response · Authors · 2025-11-29
>
> ### We appreciate the reviewer’s careful reading, but several concerns stem from a misunderstanding of our problem setting and the intended scope of our method.
> 1. This is a new task, and standard hyperedge-prediction baselines do not apply. Our work introduces a reaction-hypergraph virtual screening task that is structurally different from existing hyperedge-prediction problems. Prior models (Hyper-SAGNN, NHP, Hypergraph Contrastive Attention Networks) assume directed hyperedges or uniform/symmetric relations. Our task involves unordered, non-uniform reactant sets with chemistry-specific semantics, so existing baselines are not directly applicable.
> 2. HGNN is the only meaningful baseline for this setting. HGNN is the canonical higher-order generalization of GCNs and directly matches our reaction-hypergraph structure. It is the only baseline suitable for virtual screening. GCN, GAT, and NOCD appear only as graph-reduced comparisons to isolate the value of higher-order modeling. Using arbitrary hypergraph models designed for unrelated semantics would not be scientifically valid.
> 3. Simulated annealing (SA) is integral, not an external add-on. Our method addresses two sub-problems: (1) hyperedge classification over curated reactant sets (via ChemHGNN) and (2) exploration of unseen combinations, where the search space is exponential. SA is used precisely because combinatorial exploration cannot be solved end-to-end. It optimizes over ChemHGNN’s embedding space, leverages the zero-sum regularizer, and follows standard practice in molecular/protein design where neural potentials guide external search. It is model-aware and principled.
> 4. USPTO evaluation is appropriate for this methodological contribution. Our reaction-hypergraph design and RCNS negative sampling require reaction-center annotations, which only curated datasets like USPTO provide. The goal of this paper is to introduce a modeling pipeline, not to benchmark on noisy datasets lacking the labels needed for our method or for comparable HGNN-based approaches.
> 5. Our design philosophy. The model learns correlations at molecular, reaction, and hypergraph levels. The embedding-sum constraint reflects the assumption that reactants contribute complementary information. This constraint enables SA to optimize in embedding space before classification, forming a coherent discovery pipeline.
>
> ### Responses to the reviewer’s specific questions
> 1. “Can ChemHGNN score arbitrary reactant sets?” It can score any given set, but the challenge is finding good sets. The combinatorial space is enormous; brute-force enumeration or random sampling is infeasible. SA efficiently searches this space using the model’s embedding landscape.
> 2. “Why use the vector-sum norm instead of classifier score for SA?” The vector-sum norm is smoother and more stable for black-box combinatorial search. The classifier score is sharper and leads to unstable acceptance dynamics. The norm also aligns with our zero-sum embedding regularizer, meaning SA optimizes the model’s own learned geometry. Using classifier scores for SA is a possible variant but outside the scope of this first paper.
> 3. “How do SA hyperparameters affect performance?” More iterations improve discovery until plateauing. Temperature and cooling control exploration: higher temperature increases diversity and lower temperature speeds convergence but risks local optima.
> 4. “Is the zero-sum embedding a chemistry law?” No. It is a modeling constraint that encodes complementarity among reactants. Since training encourages valid reactions to lie in low-norm regions, SA searches effectively within this geometry. We do not claim it as a universal chemical principle.
> 5. “Why not evaluate on real-world datasets?” Our hypergraph construction and RCNS require reaction-center labels, which most real-world datasets lack.

---

### Meta-Review · Area_Chair_q7P9 · 2026-01-09

**Summary:**

Overall, the paper is viewed as technically solid and clearly argued, but with questions remaining about contribution strength, generality, and positioning relative to existing methods. More specifically, the main concerns are: (i) limited novelty -- components are known; (ii) restricted experimental scope, relying exclusively on USPTO subsets with no external datasets or real-world case studies; (iii) insufficient baseline coverage, especially the absence of more recent hypergraph or chemistry-specific methods; and (iv) ambiguity around the role of the simulated annealing “Sort Out Block,” seen as external to the core model.

**Reviewer Concerns:**

Concerns addressed:

- Task formulation: The authors clearly explained that their setting is reaction virtual screening (reactant-set feasibility) rather than product prediction or general hyperedge prediction. This explains why several suggested baselines (directed hypergraphs, reaction graph models, product-conditioned models) are not directly applicable.
- Role of simulated annealing (SA): The rebuttal clarified that SA is an integral, model-aware search component used to explore combinatorial reactant spaces.
- RCNS effectiveness: The rebuttal explained why RCNS is not expected to outperform simpler negative sampling in raw F1. Its role is in generating harder negatives, improving generalization, and preventing collapse.
- Directed hypergraphs: Authors clarified why directionality is not appropriate for a pure feasibility screening task without explicit product modeling.

Concerns remain:
- Novelty is more on engineering.
- The lack of comparisons to any more recent or alternative hypergraph operators. Dependence on USPTO alone is justified but still limits external validity.

**Reviewer Scores:**

- Reviewer v4aA: Likely unchanged.
- Reviewer WQKT: Could be a bit more positive.
- Reviewer XhcC: Unchanged or slightly improved.

---

### Decision · Program_Chairs · 2026-01-26

Reject